# An oxygen-insensitive Hif-3α isoform inhibits Wnt signaling by destabilizing the nuclear β-catenin complex

**Peng Zhang[1], Yan Bai[1], Ling Lu[2], Yun Li[2], Cunming Duan[1]\***

[1]Department of Molecular, Cellular, and Developmental Biology, University of Michigan, Ann Arbor, United States; [2]Key Laboratory of Marine Drugs, Ministry of Education and School of Medicine and Pharmacy, Ocean University of China, Qingdao, China

**Abstract** Hypoxia-inducible factors (HIFs), while best known for their roles in the hypoxic response, have oxygen-independent roles in early development with poorly defined mechanisms. Here, we report a novel Hif-3α variant, Hif-3α2, in zebrafish. Hif-3α2 lacks the bHLH, PAS, PAC, and ODD domains, and is expressed in embryonic and adult tissues independently of oxygen availability. Hif-3α2 is a nuclear protein with significant hypoxia response element (HRE)-dependent transcriptional activity. Hif-3α2 overexpression not only decreases embryonic growth and developmental timing but also causes left-right asymmetry defects. Genetic deletion of Hif-3α2 by CRISPR/Cas9 genome editing increases, while Hif-3α2 overexpression decreases, Wnt/$\beta$-catenin signaling. This action is independent of its HRE-dependent transcriptional activity. Mechanistically, Hif-3α2 binds to $\beta$-catenin and destabilizes the nuclear $\beta$-catenin complex. This mechanism is distinct from GSK3$\beta$-mediated $\beta$-catenin degradation and is conserved in humans. These findings provide new insights into the oxygen-independent actions of HIFs and uncover a novel mechanism regulating Wnt/$\beta$-catenin signaling.

**\*For correspondence:** cduan@ umich.edu

**Competing interests:** The authors declare that no competing interests exist.

## Introduction

Hypoxia-inducible factors (HIF) are evolutionarily conserved transcriptional regulators that play key roles in coordinating the cellular response to hypoxia (*Semenza, 2012*). HIFs are heterodimers, consisting of an oxygen-regulated α subunit and a stable β subunit. All three HIFα proteins contain a bHLH (basic helix-loop-helix) domain, two PAS (Per-Arnt-Sim) domains, a PAC (PAS associated C-terminal) domain, and an ODD (oxygen-dependent degradation) domain. HIF-1α and HIF-2α have two TADs (transactivation domains) (*Pugh et al., 1997*; *Tian et al., 1997*). HIF-3α has only one TAD but it has a unique LZIP (leucine zipper) domain in the C-terminal region (*Gu et al., 1998*; *Hara et al., 2001*). Under normal oxygen tension, HIFα proteins are hydroxylated at conserved proline residues in their ODD domains by PHDs (prolyl hydroxlase domain proteins). The prolyl hydroxylated HIFα proteins are recognized by the pVHL (von Hippel-Lindau protein). pVHL polyubiquitinates HIFα proteins and targets them to the proteasome for degradation (*Huang et al., 2002*; *Kageyama et al., 2004*). Under hypoxia, prolyl hydroxylation is decreased and HIFα proteins are stabilized. They then dimerize with HIFβ, translocate into the nucleus, bind to the hypoxia response elements (HREs) in the promoter regions of their target genes, and up-regulate their expression (*Semenza, 2012*; *Simon and Keith, 2008*). These HIF target genes participate in many adaptive and pathological processes such as erythropoiesis, angiogenesis, metabolic reprograming, cell-cycle regulation, and tumorigenesis (*Semenza, 2012*; *Simon and Keith, 2008*). In addition to their well-known functions in coordinating the transcriptional responses to hypoxia, HIFs have been shown to have oxygen-

**eLife digest** Proteins known as hypoxia-inducible factors (HIFs) are important in animals when the amount of oxygen in the air or water drops. These proteins switch on genes that help cells and tissues adapt to the shortage in oxygen, for example, by stimulating the production of red blood cells. Each HIF is made up of two subunits called α and β that only bind to each other when the oxygen levels drop. This two-subunit complex, or 'dimer', then activates a set of genes by binding to a stretch of DNA known as the hypoxia response element. HIFs also play important roles in many different stages of animal development. There are many different HIF proteins that are each present at different developmental stages; this has made them difficult to study.

Zhang et al. have found a new form of HIF-3α in zebrafish – called Hif-3α2. The experiments show that this α subunit is not regulated by oxygen, but may still be able to activate genes that have the hypoxia response element. When Hif-3α2 was injected into zebrafish embryos, the body pattern that is normally set up in embryogenesis was disrupted. Further experiments revealed that Hif-3α2 regulates embryo development by destabilizing a protein called β-catenin. This inhibits a cell communication system called Wnt/β-catenin signaling. Zhang et al. also show that the two distinct activities of Hif-3α2 – binding to the hypoxia response element and destabilizing β-catenin – involve two different regions of the protein.

Together, Zhang et al.'s findings show that zebrafish Hif-3α2 combines some conventional features of HIF proteins with a unique developmental role. It is likely that human Hif-3α may also work in a similar way, so future studies will focus on understanding the molecular mechanisms responsible for these distinct roles.

independent roles in early development in mammals, frogs, fish, and invertebrates (*Barriga et al., 2013*; *Dunwoodie, 2009*; *Simon and Keith, 2008*). The mechanisms underlying these oxygen-independent roles of HIFs, however, are not well understood.

The HIFα genes are subjected to sophisticated post-transcriptional regulation with that of HIF-3α being the most complicated (*Prabhakar and Semenza, 2012*). The human *HIF-3α* gene, for example, has 19 predicted variants that result from the use of different promoters, different transcription initiation sites, and alternative splicing (*Duan, 2015*). Eight of them have been experimentally shown to encode proteins (*Heikkila et al., 2011*; *Maynard et al., 2003*; *Pasanen et al., 2010*). The mouse *Hif-3α* locus also gives rise to several different variants, resulting in the full-length HIF-3α, NEPAS (neonatal and embryonic PAS), IPAS (inhibitory PAS) and possibly others (*Gu et al., 1998*; *Makino et al., 2001*; *Yamashita et al., 2008*). These isoforms are often expressed in different tissues, at different developmental stages, and are differentially regulated. They have distinct or even opposite functions when tested by overexpression approaches (*Duan, 2015*). For instance, while human HIF-3α1, the full-length human HIF-3α, can stimulate HRE-dependent reporter construct activity and up-regulate unique target genes (*Gu et al., 1998*; *Zhang et al., 2014*), human HIF-3α4 isoform, a shorter isoform that lacks the TAD domain, inhibits the activity of HIF-1α and HIF-2α (*Maynard et al., 2005*; *Maynard et al., 2007*). Similarly, IPAS was shown to inhibit HIF-1α activity (*Makino et al., 2001*), while NEPAS has weak transcriptional activity and is thought to inhibit HIF-1/2α activity by competing for the common HIFβ in cells with limited amounts of HIFβ (*Yamashita et al., 2008*). The existence of such a large array of HIF-3α variants has posed enormous challenges to studying HIF-3 biology. While the conventional gene knockout technology has been used to knockout the NEPAS/HIF-3α/IPAS in mice (*Yamashita et al., 2008*), the interpretation of the results is not straightforward because multiple isoforms are deleted. The new CRISPR/Cas9 genome editing technology makes it possible to address this problem.

Wnts are secreted glycoproteins that play crucial roles in cell fate specification, body axis determination, cell proliferation, and cell migration during embryogenesis (*Clevers and Nusse, 2012*; *MacDonald et al., 2009*). The Wnt signaling pathway also regulates stem cell renewal and adult tissue homeostasis. Aberrant expression and/or activation in Wnt signaling leads to many human diseases such as birth defects, cancer, and degenerative disorders (*Clevers and Nusse, 2012*; *MacDonald et al., 2009*). In the absence of Wnt ligands, the transcriptional co-activator β-catenin is

phosphorylated in the cytoplasm by a protein complex consisting of APC, CK1, Axin, and GSK3β. This leads to β-catenin recognition by the ubiquitin ligase β-TrCP. β-TrCP binds to the N-terminal region of β-catenin in a phosphorylation-dependent manner and promotes β-catenin degradation. The binding of a Wnt ligand to Frizzled and co-receptors inhibits the phosphorylation and degradation of β-catenin. The stabilized β-catenin accumulates and translocates into the nucleus to form complexes with TCF/LEF, and thereby activates target gene expression (*Clevers and Nusse, 2012*; *MacDonald et al., 2009*). In addition to the canonical pathway, Wnt also regulates planar cell polarity and Akt/mTOR through non-canonical pathways (*Clevers and Nusse, 2012*; *MacDonald et al., 2009*)

We have recently shown that the full-length zebrafish Hif-3α is an oxygen-dependent transcription factor and that it activates a transcriptional program distinct from that of Hif-1α in zebrafish embryos under hypoxia (*Zhang et al., 2014*). In this study, we have identified a novel zebrafish Hif-3α spliced variant, termed Hif-3 isoform 2 (Hif-3α2). Hif-3α2 is an oxygen-insensitive nuclear protein. Despite its lack of the bHLH and PAS domains, Hif-3α2 has HRE-dependent transcriptional activity. We investigated the in vivo role of Hif-3α2 using transgenesis and CRISPR/Cas9-mediated gene editing. Our results suggest that Hif-3α2 inhibits canonical Wnt signaling by binding to β-catenin and destabilizing the nuclear β-catenin complex. This action is independent of its HRE-dependent transcriptional activity and is evolutionarily conserved.

## Results

### Hif-3α2 is a novel oxygen-insensitive Hif-3a isoform

RT-PCR analysis of zebrafish embryo RNA detected two major transcripts (*Figure 1A*). In addition to the previously reported full-length *hif-3α* transcript (*Zhang et al., 2012*), there is a short transcript. These transcripts are referred to as *hif-3α* isoforms 1 and 2, respectively (*hif-3α1 and -3α2*) hereafter. The complete sequence of the *hif-3α2* transcript (893 bp) was determined by 5' and 3' RACE (GenBank access number KR338972). It lacks exons 3–10 and a portion of exon 2 (*Figure 1B*). The predicted ORF is 513 bp, encoding a protein of 170 amino acids with a predicted size of 19 kDa. The encoded protein contains the TAD and LZIP domains but lacks the bHLH, PAS, PAC, and ODD domains (*Figure 1B*). During early development, *hif-3α2* mRNA levels peaked at 9–16 hr post fertilization (hpf), decreased after 20 hpf, and increased again in the larval stage (*Figure 1C*). In the adult stage, the highest levels of *hif-3α2* mRNA were found in the kidney, followed by ovary, spleen, testis, intestine, brain, and heart. The lowest levels were detected in the liver (*Figure 1D*). Its levels were lower compared with those of *hif-3α1* mRNA (*Figure 1—figure supplemental 1A*).

Because Hif-3α2 lacks a complete ODD domain, we speculated that its protein stability might not be regulated by oxygen tension. To test this idea, capped mRNA encoding GFP, Hif-3α1-GFP, and Hif-3α2-GFP was injected into 3 separate groups of zebrafish embryos. The embryos were raised to 6 hpf under normoxia. As previously reported, Hif-3α1-GFP was rapidly degraded under normoxia and no GFP signal was observed in this group (*Zhang et al., 2014*). In contrast, strong signal was observed in Hif-3α2-GFP and GFP mRNA injected embryos (*Figure 1E*). The Hif-3α2-GFP signal was observed in the nucleus, while the GFP protein was present in the cytoplasm and nucleus (*Figure 1E*). Western blotting confirmed these results (*Figure 1F*). The levels of endogenous Hif-3α2 protein were examined using a validated Hif-3α antibody (*Zhang et al., 2012*). This antibody detected both Hif-3α1 and Hif-3α2 in HEK293T cells transfected with a GFP tagged construct (*Figure 1—figure supplement 1B*). Western blotting results showed comparable levels of Hif-3α2 in zebrafish embryos raised in normoxic and hypoxic water (*Figure 1G*), while hypoxia induced a robust increase in Hif-3α1 levels (*Figure 1G*). Likewise, 6 hr, 12 hr, and 24 hr hypoxia treatments did not change Hif-3α2 levels in adult zebrafish (*Figure 1H*). These results suggest that unlike Hif-3α1, Hif-3α2 expression is not regulated by oxygen availability.

### Hif-3α2 is a nuclear protein with HRE-dependent transcriptional activity

Co-expression of Hif-3α2 with p2.1, a HRE reporter construct (*Semenza et al., 1996*), resulted in a highly significant, 11.3-fold increase in the reporter activity in HEK293 cells (*Figure 2A*). This activity was abolished when the HRE was mutated (i.e., p2.4 in *Figure 2A*). In comparison, zebrafish Hif-3α1 and Hif-1α caused a 27.4-fold and 14.5-fold increase, respectively (*Figure 2A*). Similar results were

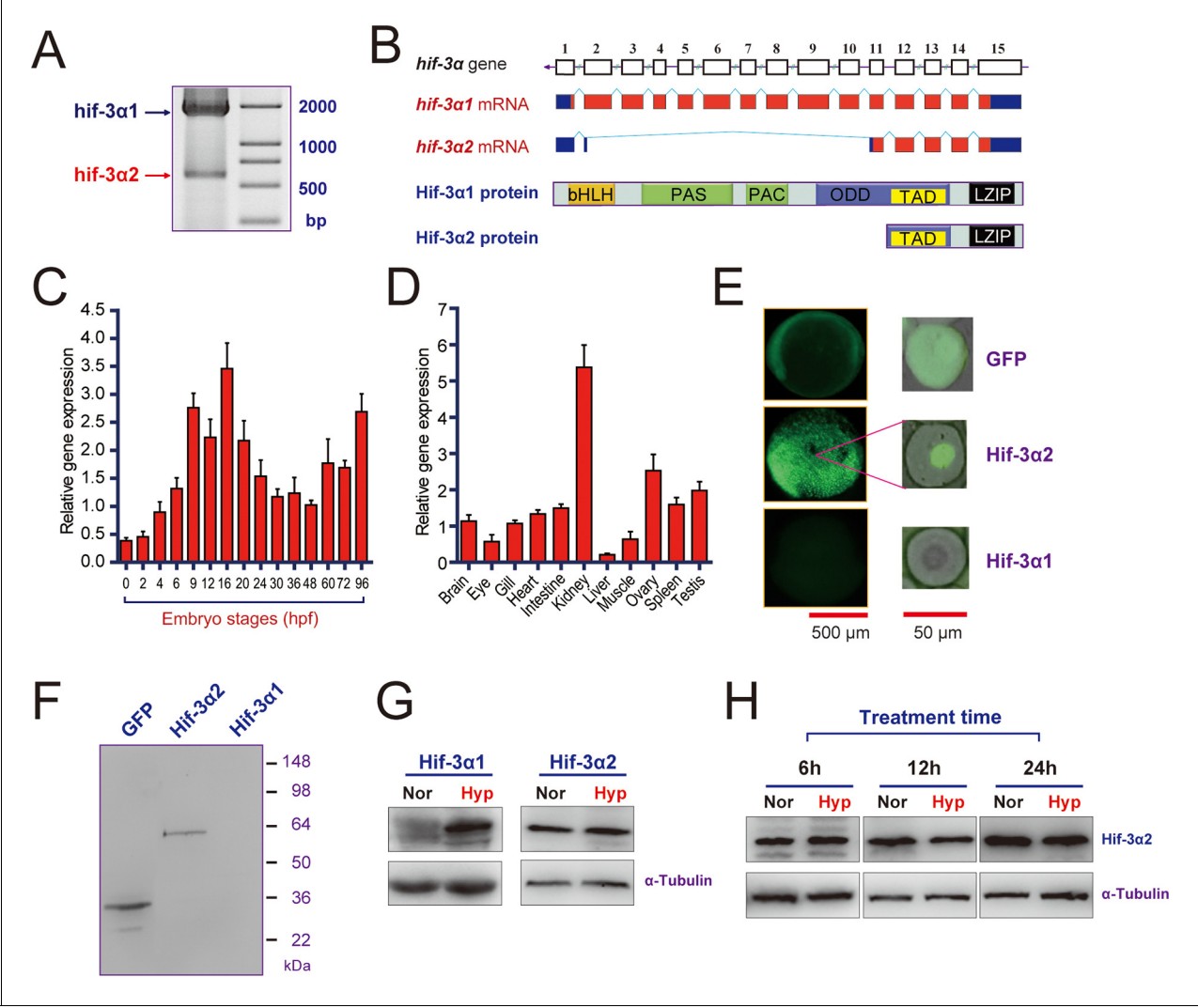

**Figure 1.** Hif-3α2 is an oxygen-insensitive Hif-3α isoform resulting from alternative splicing. (A) Hif-3α isoform 1 (Hif-3α1) and isoform 2 (Hif-3α2) mRNA expression. RNA isolated from zebrafish embryos was analyzed by RT-PCR. (B) Schematic illustration of the *hif-3α* gene (top), Hif-3α1 and Hif-3α2 mRNAs (middle), and proteins (bottom). (C,D) Hif-3α2 expression in early development (C) and in adult tissues (D). The Hif-3α2 mRNA levels were measured by qRT-PCR and normalized by *β-actin* levels. Values are means $\pm$S.E. (n = 3). (E,F) Hif-3α1-EGFP but not Hif-3α2-EGFP is degraded under normoxia in vivo. Capped mRNA encoding EGFP, Hif-3α1-EGFP, and Hif-3α2-EGFP was injected into zebrafish embryos. The embryos were raised to 6 hpf under normoxia and observed under fluorescence microscopy (E) or analyzed by Western blotting using an anti-GFP antibody (F). (G) 6-hpf wild-type embryos were transferred to hypoxic (Hyp) or normoxic water (Nor) for 24 hr and analyzed by Western blotting using a specific Hif-3α antibody. (H) Adult fish were subjected to hypoxia treatments for the indicated time period and analyzed by Western blotting.

The following figure supplement is available for figure 1:

**Figure supplement 1.** Hif-3α mRNA expression in adult and antibody validation.

obtained in HeLa cells (*Figure 2—figure supplemental 1B*). When tagged with GFP and transfected into human cells, Hif-3α2-GFP signal was seen in the nucleus in cultured cells and in zebrafish embryos (*Figure 2C and E*). Deletion of the TAD abolished the transcriptional activity, while it only slightly reduced the nuclear GFP signal intensity (*Figure 2B–C*). Deletion of the LZIP domain had no effect on the nuclear localization but increased the transcriptional activity (*Figure 2B–D*). The expression levels of the ΔLZIP mutant were higher than that of Hif-3α and ΔTAD (*Figure 2—figure supplemental 1A*). Overexpression of Hif-3α2 in zebrafish embryos resulted in significant increases in the mRNA levels of *igfbp-1a*, *mlp3c*, and *redd1* (*Figure 2F–H*). These genes are known Hif-1α and Hif-

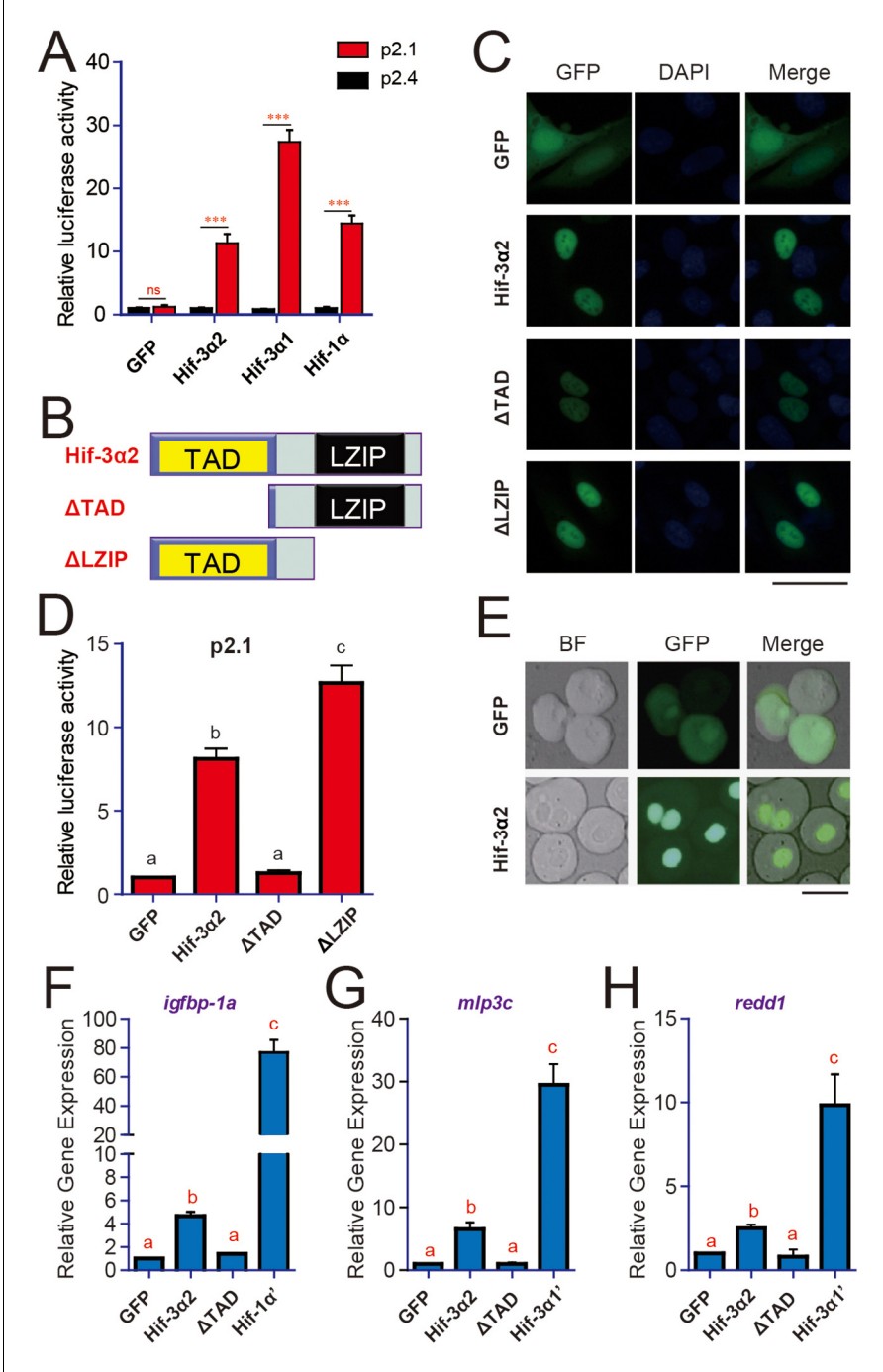

**Figure 2.** Hif-3α2 is a nuclear protein and has HRE-dependent transcriptional activity. (**A**) HRE-dependent transcriptional activity. HEK293 cells were transfected with the indicated plasmid together with 100 ng p2.1 (red) or p2.4 plasmid (black). The results are normalized and expressed as fold change over the GFP p2.4 group. Values are means ± S.E. (n = 3). ***p < 0.001. (**B**) Schematic diagram of the Hif-3α2 truncation mutants tested. (**C**) U2OS cells were transfected with the constructs shown in (**B**). The GFP signal was visualized 24 hr after transfection (left panels). Cells were counterstained with DAPI (middle panels). Merged views are shown in the right panels. Scale bar = 50 μm. (**D**) HRE-dependent transcriptional activity of Hif-3α2 mutants. Values are means ± S.E. (n = 3). Groups labeled with different letters are significantly different from each other (*P* < 0.05). (**E**) Nuclear localization of Hif-3α2-EGFP in zebrafish embryo cells. Embryos injected with EGFP (600 pg) or Hif-3α2-EGFP capped mRNA (600 pg) were raised to 6 hpf under normoxia. The cells were dispersed and observed under bright field (BF) and GFP fluorescence microscopy. Scale bar = 50 μm. (**F–H**) Effects of Hif-3α2 (600 pg), Hif-3α1' (stabilized Hif-3α1, 800 pg),
*Figure 2 continued on next page*

*Figure 2 continued*

and Hif-1α′ (stabilized Hif-1α, 800 pg) on endogenous gene expression. Embryos injected with the indicated capped mRNA were raised to 12 hpf under normoxia. The mRNA levels of the indicated genes were determined by qRT-PCR and normalized by the *β-actin* levels. Values are means $\pm$ S.E. (n = 3). Groups labeled with different letters are significantly different from each other ($P < 0.05$).

The following figure supplement is available for figure 2:

**Figure supplement 1.** Hif-3α2′s transcriptional activity differs from that of Hif-1α and Hif-3α1.

3α target genes in zebrafish (*Feng et al., 2012*; *Kajimura et al., 2006*; *Kamei et al., 2008*). The ΔTAD mutant had no such effect (*Figure 2F–H*). The magnitude of induction by Hif-3α2, however, was much lower compared to that of Hif-3α1 or Hif-1α (*Figure 2F–H*). While Hif-1α increased the expression of its target gene *vegfAb*, Hif-3α2 had no such effect (*Figure 2—figure supplemental 1C*). Likewise, Hif-3α1 but not Hif-3α2 increased the mRNA levels of *zp3v2* and *sqrdl* (*Figure 2—figure supplemental 1D–E*). These data suggest that Hif-3α2 is a nuclear protein capable of activating HRE-dependent gene expression in vitro and in vivo. They also suggest that Hif-3α2 either does not activate the target genes to the same degree or has target genes differ from that of full-length Hif-3α1.

## Forced expression of Hif-3α2 but not Hif-3α1 leads to left-right (LR) asymmetry defects

The discovery of Hif-3α2 has raised the possibility that it may play a role in early development in an $O_2$-independent manner. Overexpression of Hif-3α2 resulted in reduced body growth and developmental timing in zebrafish embryos (*Figure 3—figure supplemental 1A–B*). This effect is similar to what has been reported for the full-length Hif-3α1 (*Zhang et al., 2014*). This action of Hif-3α2 appears to require its transcriptional activity because overexpression of the ΔTAD mutant did not change body size or somite number, while the ΔLZIP mutant did (*Figure 3—figure supplemental 1A–B*). While GFP mRNA-injected or ΔTAD mRNA-injected embryos were mostly morphologically normal, many Hif-3α2- and ΔLZIP-expressing embryos exhibited morphological abnormality, (*Figure 3—figure supplemental 1C–D*). Approximately 84% of the Hif-3α2 mRNA-injected embryos had partial or complete loss of *myod1*-mRNA expressing somites on one or both sides of the body (*Figure 3A*). Similar results were found with *myog* mRNA expression (*Figure 3B*). The expression patterns of two asymmetric genes, *spaw* and *lefty2* (*lft2*) (*Amack et al., 2007*), were examined. While the *spaw* expression was detected on the left side of the body plan in all GFP mRNA-injected embryos, its expression was detected on the right side, bilaterally, or was completely lost in ~58% of the Hif-3α2-expressing embryos (*Figure 3C*). Similar changes were found with *lft2* expression (*Figure 3D*). During zebrafish development, the shape of the heart changes from a tube-like structure in the middle to a left looped structure (*Stainier, 2001*). The cardiac tube remained in the middle or even looped to the right side in half of the Hif-3α2 expressing embryos (*Figure 3E*). No such phenotype was observed in Hif-3α1 mRNA-injected embryos (*Figure 3E*). The effect of Hif-3α2 expression was further examined using LiPan fish, a transgenic zebrafish line that has liver-specific expression of DsRed RFP and pancreas-specific expression of GFP (*Korzh et al., 2008*). In the control embryos, the liver was located on the left and pancreas on the right side of the body plan in all embryos (*Figure 3F*). In the Hif-3α2 group, the liver was found on the right side or in the midline in many embryos (28.5%) (*Figure 3F*). These data suggest that overexpression of Hif-3α2 but not the full-length Hif-3α1 impairs LR axis development.

## Forced expression of Hif-3α2 but not Hif-3α1 inhibits Wnt/β-catenin signaling and impairs Kupffer's vesicle development

Kupffer's vesicle (KV), a transient embryonic organ, plays a key role in establishing the LR asymmetry axis in zebrafish (*Essner et al., 2005*). The rotating cilia of KV establish a counterclockwise fluid flow and promote intracellular $Ca^{2+}$ elevation in cells localized on the left side of KV, which in turn stimulates the processing and/or secretion of the Nodal-related ligand gene *spaw* on the left side of KV (*Husken and Carl, 2013*). KV is derived from the dorsal forerunner cells (DFCs). DFCs emerge at

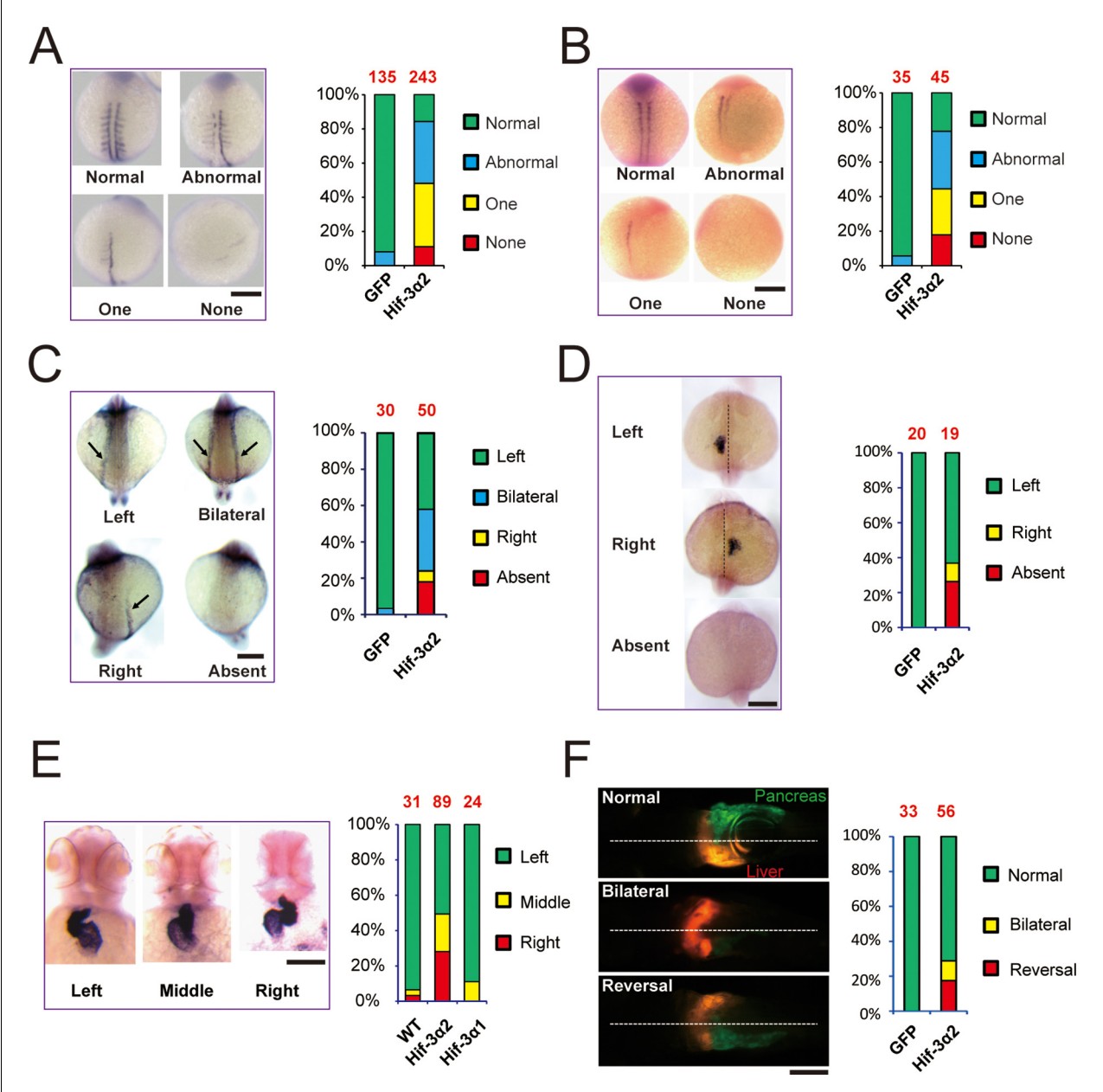

**Figure 3.** Forced expression of Hif-3α2 causes LR asymmetry defects. Embryos injected with GFP or Hif-3α2 capped mRNA (600 pg) were raised and analyzed by in situ hybridization. (A,B) Somite development was visualized by *myod1* (A) and *myog* (B) mRNA expression at 12–14 hpf. The embryos were scored based on the criteria shown in the left panel. The quantification results are shown in the right. The total embryo number is shown on the top of each column. (C) Altered expression of *spaw* (C) mRNA at 18 hpf and *lft2* mRNA (D) at 20 hpf. (E) The cardiac tube looping was visualized by *cmlc2* mRNA expression at 48 hpf. Representative views are shown in the left panel. The quantification results are shown in the right panel. Hif-3α1 (stabilized Hif-3α1, 800 pg) injected embryos were used as controls. (F) Changes in liver and pancreas location. Hif-3α2 capped mRNA was injected into LiPan transgenic embryos. Liver (red) and pancreas (green) location was examined at 96 hpf.

The following figure supplement is available for figure 3:

**Figure supplement 1.** Forced expression of Hif-3α2 slows down embryonic growth and developmental timing and causes morphological abnormality.

mid-gastrulation and migrate collectively to the vegetal pole. At the tail bud, they form a rosette structure. The lumen and cilia are then developed to form KV (*Matsui and Bessho, 2012*). We investigated whether Hif-3α2 overexpression alters LR asymmetry axis by affecting DFCs and/or KV development.

In situ hybridization and qRT-PCR analyses using the DFC marker gene *sox17* indicated that Hif-3α2 overexpression caused a modest increase in DFC cluster size and total mRNA levels, but these increases were not statistically significant (*Figure 4A and C*). There was a slight decrease in the number of migrating DFCs (*Figure 4B*). We analyzed KV forming using *charon*, which is expressed in KV lumen epithelia and surrounding cells (*Hashimoto et al., 2004*). While wild-type embryos had normally formed KVs, many Hif-3α2 mRNA-injected embryos had a partially formed KVs (48%) or no KV (20%) (*Figure 4D*). This action is specific to Hif-3α2 because Hif-3α1 overexpression had no such

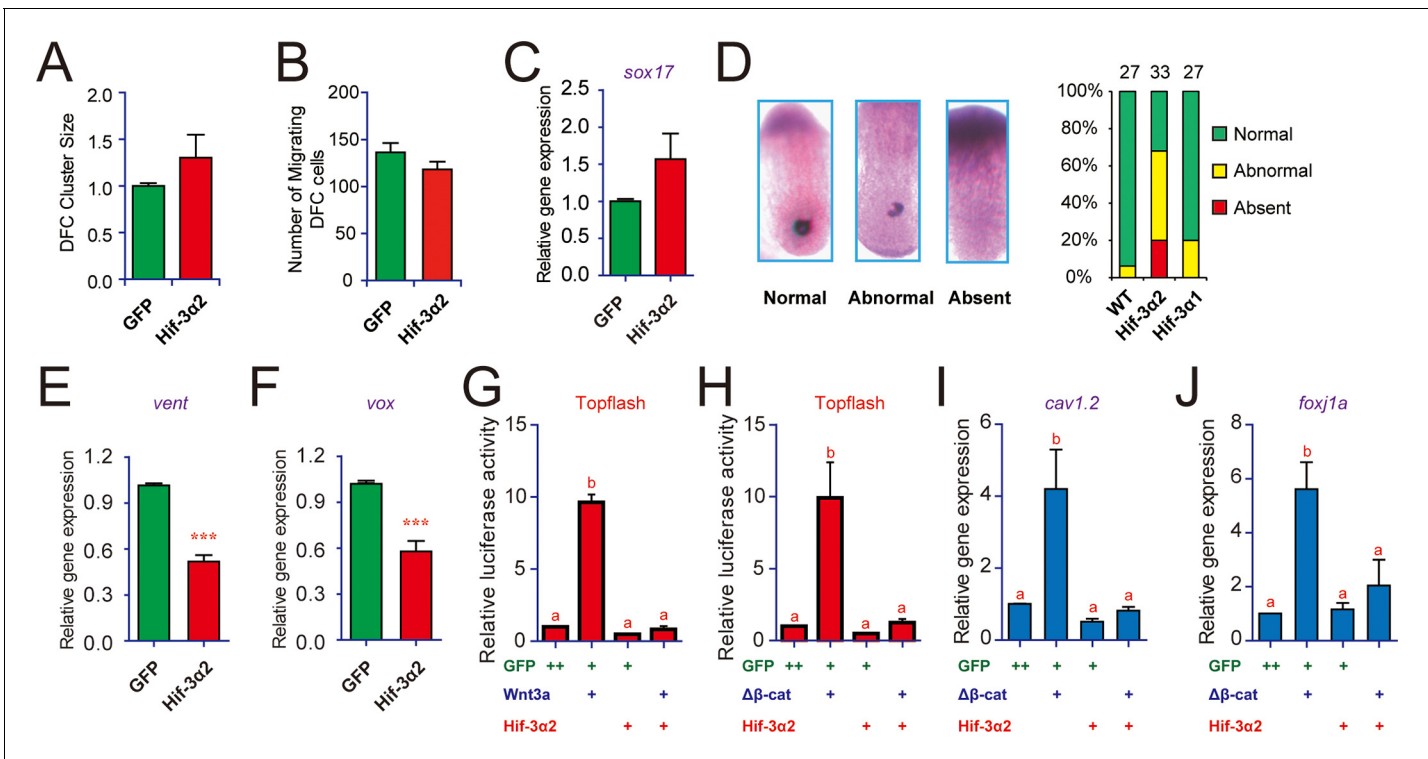

**Figure 4.** Forced expression of Hif-3α2 impairs Kupffer's vesicle (KV) development and inhibits Wnt/β-catenin signaling. (A,B) Lack of effect on dorsal forerunner cell (DFC) development. The DFC cluster size (A) was determined by measuring *sox17* mRNA expression domain in the Hif-3α2 mRNA injected embryos at 8 hpf (n = 22) using ImageJ. The values were normalized by those of the GFP mRNA injected embryos (n = 19). The number of migrating DFCs were quantified and shown in (B). (C) The *sox17* mRNA levels were determined by qRT-PCR and normalized by the *β-actin* levels. Values are means ±S.E. (n = 3). No significant difference is detected. (D) Effects on KV development. KV was visualized by *charon* mRNA expression at 12 hpf. The embryos were scored based on the criteria shown in the left panel. The quantification results are shown in the right. (E,F) Effects on Wnt target gene expression. Embryos injected with the indicated mRNA were raised to 9 hpf and *vent* (E) and *vox* (F) mRNA levels were determined by qRT-PCR and normalized to the *β-actin* levels. Values are means ± S.E. (n = 3). ***p < 0.001. (G,H) Inhibition of Wnt3a (G) and β-cateninΔN (H) activity. Embryos injected with Topflash plasmid DNA (90 pg) together with the 30 pg Wnt3a or 100 pg β-cateninΔN and 600 pg Hif-3α2 capped mRNA were raised to 9 hpf and luciferase activity was measured. Values are means ± S.E. (n = 3). Groups labeled with different letters are significantly different from each other (*P* < 0.05). (I,J) Inhibition of β-cateninΔN-induced *cav1.2* (I) and *foxj1a* (J) expression.qRT-PCR was performed and analyzed as described above.

The following figure supplements are available for figure 4:

**Figure supplement 1.** Forced expression of Hif-3α2 does not change Fgf, Hedgehog, Notch, Akt, and Erk signaling.

**Figure supplement 2.** Effects of Hif-3α2, Hif-3α1' (a stabilized Hif-3α1), and Hif-1α' (a stabilized Hif-1α) on Wnt3a- (A) and β-cateninΔN-induced (B) Topflash activity.

effect (*Figure 4D*). Taken together, the results suggest that Hif-3α2 overexpression causes LR asymmetry defects by impairing KV formation.

Several signaling pathways, including Wnt/β-catenin, Fgf, Notch, Hedgehog, and Akt, have been implicated in KV organogenesis (*Husken and Carl, 2013*; *Matsui and Bessho, 2012*; *Roussigne et al., 2012*). qRT-PCR assays showed that Hif-3α2 overexpression resulted in a significant reduction in the mRNA expression of *vent* and *vox* (*Figure 4E, F*), two genes acting downstream of the canonical Wnt/β-catenin pathway. No significant changes were detected in the mRNA expression of the Fgf target gene *dusp6*, the Hedgehog target gene *gli1*, or the Notch target gene *her4* (*Figure 4—figure supplemental 1A–C*). No marked changes were observed in the phospho-Akt and phospho-Erk levels (*Figure 4—figure supplemental 1D–E*). Topflash, a Luciferase reporter construct containing TCF binding sites was used to directly test whether Hif-3α2 inhibits Wnt/β-catenin signaling (*Flowers et al., 2012*). Injection of Wnt3a mRNA into zebrafish embryos resulted in a robust induction in Topflash reporter activity (*Figure 4G*). This induction was abolished by Hif-3α2 co-injection (*Figure 4G*). Co-injection of Hif-3α2 also abolished the induction of Topflash activity by β-cateninΔN (*Figure 4H*). β-cateninΔN is a constitutively active β-catenin mutant lacking part of the N-terminal sequence (*Xiong et al., 2006*). This action is specific to Hif-3α2 because the full-length Hif-3α1 did not inhibit Wnt3a- or β-cateninΔN-induced Topflash activity (*Figure 4—figure supplemental 2A–B*). Hif-1α had a modest but statistically significant effect on Wnt3a-induced Topflash activity (*Figure 4—figure supplemental 2A–B*). Co-injection of Hif-3α2 also abolished the Wnt3a- and β-cateninΔN-induced dorsalization phenotype (*Figure 4—figure supplemental 2C*). Wnt/β-catenin signaling has been shown to regulate KV development and LR asymmetry by stimulating the expression of the ciliogenic transcription factor *foxj1a* (*Caron et al., 2012*) and the calcium channel *cav1.2* (*Muntean et al., 2014*). If Hif-3α2 impairs KV development via inhibition of Wnt/β-catenin signaling, then expression of Hif-3α2 should block Wnt/β-catenin-induced *foxj1a* and *cav1.2* gene expression. Indeed, co-injection of Hif-3α2 mRNA abolished the β-cateninΔN induction of *foxj1a* and *cav1.2* gene expression (*Figure 4I–J*). These results suggest that forced expression of Hif-3α2 but not the full-length Hif-3α1 inhibits Wnt/β-catenin signaling and impairs Kupffer's vesicle development.

## CRISPR/Cas9-mediated ablation of Hif-3α2 increases Wnt/β-catenin signaling

To determine whether endogenous Hif-3α2 functions as a negative regulator of the Wnt/β-catenin signaling pathway, we generated *hif-3α-/-* mutants using a CRISPR/Cas9-mediated approach targeting the Hif-3α2 ATG site in exon 11 (*Figure 5A*). We identified several alleles. The *hif-3αΔ42* allele has a 42 bp deletion surrounding the Hif-3α2 ATG site (*Figure 5A–B*). RT-PCR assays showed that the expression of Hif-3α1 mRNA was not affected in the mutant embryos (*Figure 5C*). The successful ablation of Hif-3α2 protein was confirmed by Western blotting (*Figure 5D*). We next cloned and sequenced the Hif-3α1 cDNA from the mutant embryos. The Hif-3α1Δ42 has a 14 aa deletion (residues 456 to 469) located in the N-terminal end of the TAD (*Figure 5—figure supplemental 1A*). When its transcriptional activity was compared with that of the wild type Hif-3α1, Hif-3α1Δ42 had comparable activity (*Figure 5—figure supplemental 1B*). These data suggest that the expression and functionality of Hif-3α1 protein remain largely unchanged in the *hif-3αΔ42* line.

We next investigated whether loss of Hif-3α2 alters Wnt/β-catenin signaling activity. Compared with the wild-type embryos, the Topflash reporter activity was significantly elevated in the *hif-3αΔ42* mutant embryos at 9 hpf (*Figure 5E*). Likewise, the mRNA levels of *vent* and *vox* were elevated in *hif-3αΔ42* embryos at 9 hpf (*Figure 5F–G*). Similar results were obtained using the *hif-3αΔ20* allele, which has a 20 bp deletion covering the Hif-3α2 ATG site. It is predicted to be a *hif-3α2* null mutant and encode a truncated Hif-3α1 protein lacking the TAD and LZIP domains (*Figure 5—figure supplemental 2A and B*). The truncated protein also contains 8 incorrect aa (residues 455-462). Compared to wild type embryos, a significant increase in Topflash reporter activity was found in the *hif-3αΔ20* mutant line (*Figure 5—figure supplemental 2C*). These results indicate that endogenous Hif-3α2 negatively regulates the Wnt/β-catenin signaling.

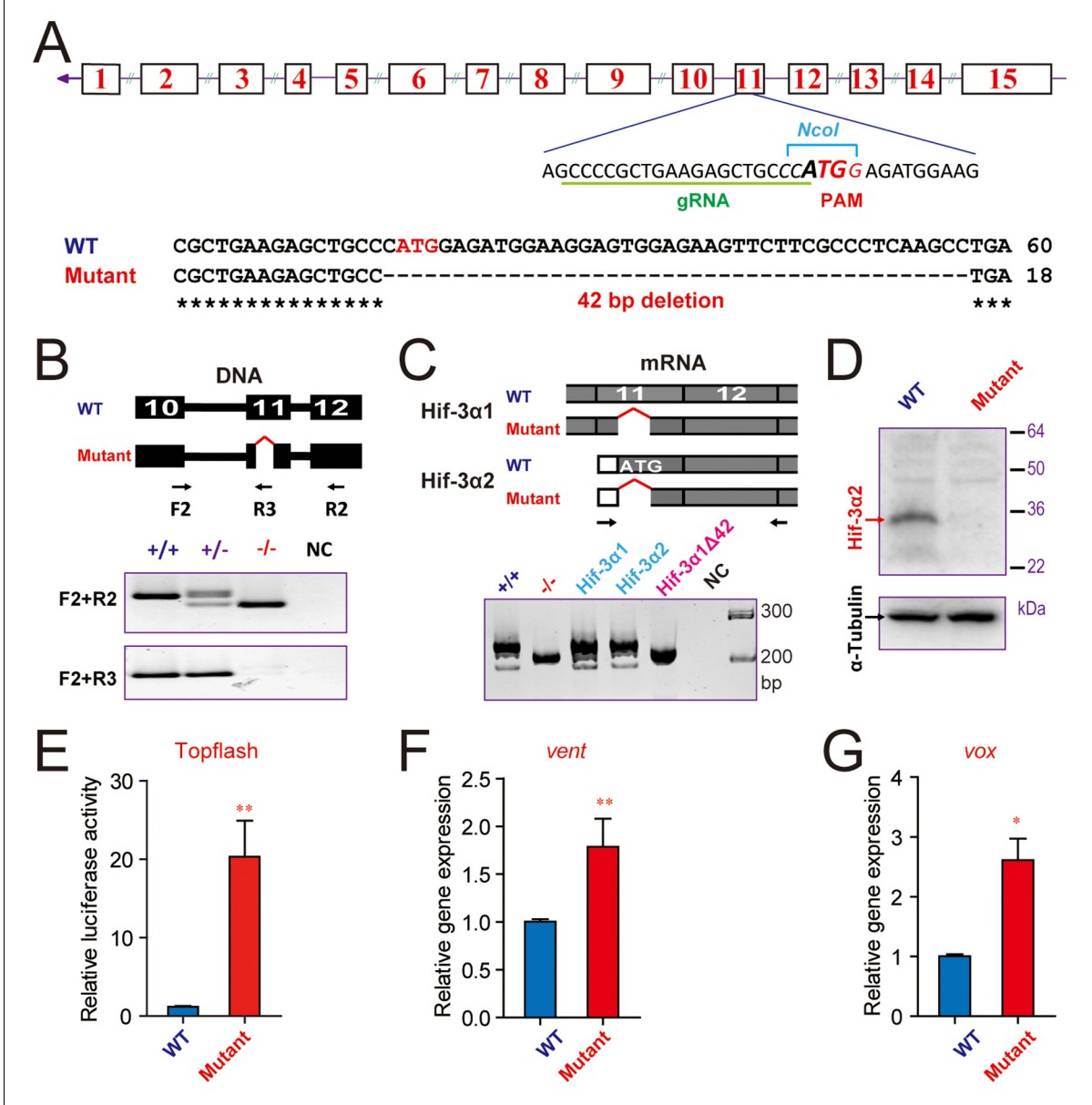

**Figure 5.** CRISPR/Cas9-mediated deletion of Hif-3α2 increases Wnt/β-catenin signaling. (**A**) Top: Design of the gRNA used to target the Hif-3α2 ATG site (bold letters) in exon 11. The gRNA targeting sequence is underlined (green). The PAM motif is labeled in red, and the NcoI digestion site is indicated in italic letters. Bottom: Sequencing results of the *hif-3α* locus in the wild type and homozygous mutant fish. (**B**) Top: Location of the 42 bp deletion and PCR primers used for genotyping. Middle and Bottom: Genotyping results using two PCR-based assays. Water was used as negative control (NC). (**C**) The expression of Hif-3α1 mRNA expression is not affected. Top: PCR primers used for detecting Hif-3α2 and Hif-3α1 mRNAs. Bottom left panel: RT-PCR results. This primer set amplified Hif-3α2 and Hif-3α1 mRNAs in the wild type (+/+) and Hif-3α1 mRNA (albeit smaller size) in the *hif-3αΔ42* mutant (-/-) embryos. Cloned Hif-3α1, Hif-3α2, and Hif-3α1Δ42 DNA were used as controls. (**D**) Western blotting results indicating the ablation of Hif-3α2 protein in the *hif-3αΔ42* mutant embryos. 24 hpf *hif-3αΔ42* mutant embryos and wild-type siblings were analyzed. (**E**) Elevated Wnt/β-catenin signaling activity. Wild-type and *hif-3αΔ42* mutant embryos were injected with 90 pg Topflash plasmid DNA and luciferase activity was measured at 9 hpf. (**F,G**) The mRNA levels of *vent* (**F**) *vox* (**G**) were measured in mutant embryos at 9 hpf and presented as relative values to the wild type controls. Values are means +S.E. (n = 3). * and **p < 0.05 and 0.01.

The following figure supplements are available for figure 5:

**Figure supplement 1.** The 14-amino acid deletion does not alter HIF-3α1 transcriptional activity.

**Figure supplement 2.** CRISPR/Cas9-mediated mutation of Hif-3α1/2 increases Wnt/β-catenin signaling.

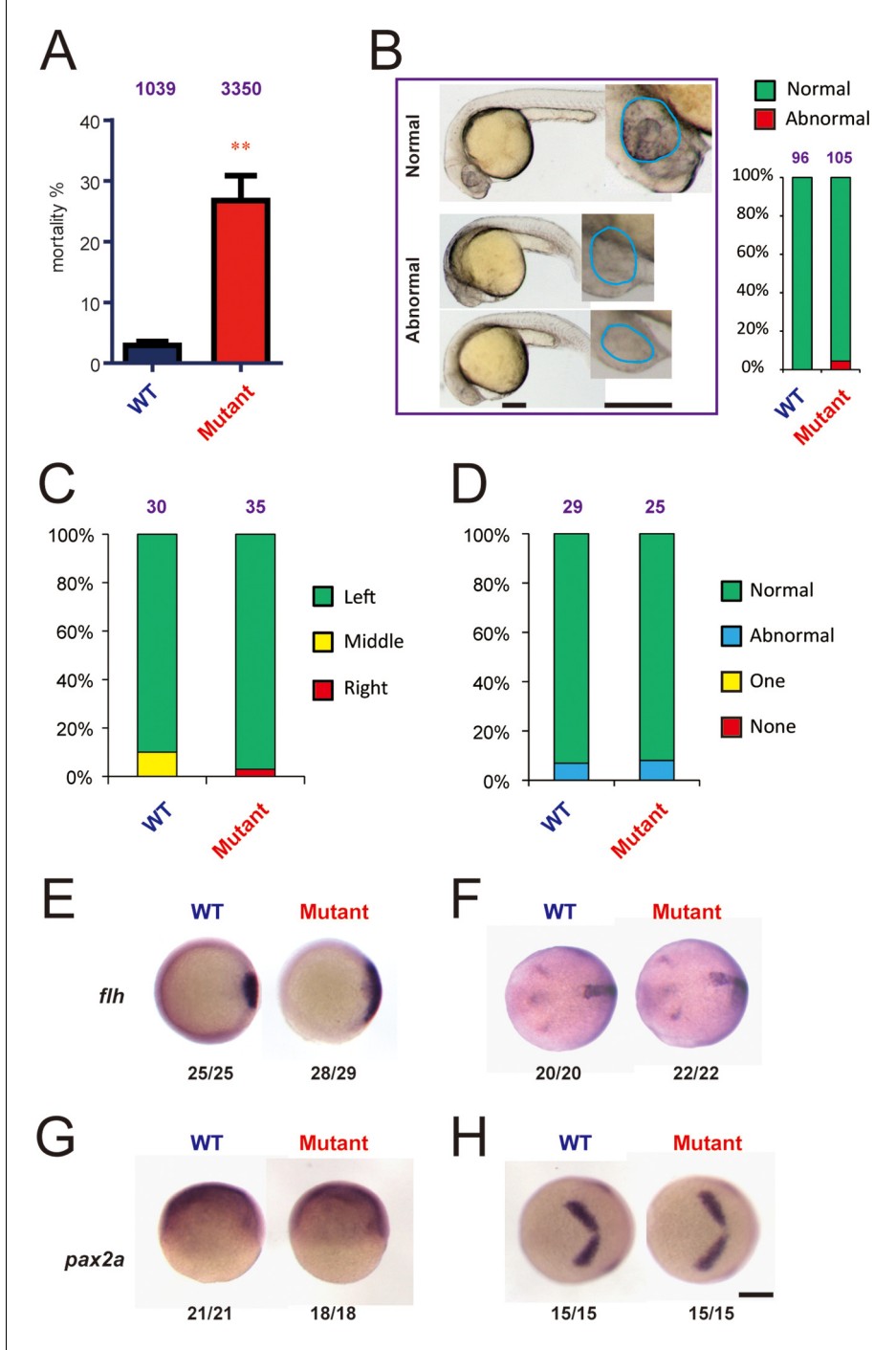

**Figure 6.** Genetic deletion of Hif-3α2 increases embryo mortality and transient Wnt/β-catenin signaling phenotype. (**A**) Increased embryo mortality. Wild-type and F3 homozygous *hif-3αΔ42* mutant embryos were raised under normoxia. The number of dead embryo was determined at 9 hpf and is shown as percentage of total embryos. The total embryo number is shown on the top of each column. (**B**) Morphology of the surviving embryos at 24 hpf. The total embryo number is shown on the top of each column. (**C,D**) The cardiac tube looping (**C**) and somite development (**D**) were examined and quantified at 48 and 12–14 hpf. (**E-H**) Expression of *flh* and *pax2.a* mRNA in wide type and *hif-3αΔ42* mutant embryos at 6 hpf (**E,G**) and 10 hpf (**F,H**). The frequency of embryos with the indicated expression patterns is shown at the bottom of each panel. Scale bar = 200 μm.

The following figure supplement is available for figure 6:

*Figure 6 continued on next page*

*Figure 6 continued*
**Figure supplement 1.** The *hif-3α∆42* line had higher mortality rate.

## Genetic hif-3α mutation exhibits transient phenotypes while knockdown of Hif-3α2 results in profound phenotypes

A most notable phenotype observed in the *hif-3α∆42* mutant line was elevated embryo mortality (*Figure 6A*). Many *hif-3α∆42* embryos died before 6–9 hpf. The mortality rate, although varied considerably among different pairs of mutant fish and between different breeding cycles in the same pair (*Figure 6—figure supplement 1*), was significantly higher than that of the wild-type controls (*Figure 6A*). The hif-3α∆20 mutant line also had significantly elevated mortality rate (*Figure 5—figure supplemental 2D*). Although two out of 105 of the surviving *hif-3α∆42* embryos had reduced forebrain and smaller eyes, most of the surviving *hif-3α∆42* mutant embryos looked morphologically normal and continued to develop (*Figure 6B*). No difference was detected in the somite development and cardiac looping between the mutant and wild-type embryos (*Figure 6C–D*). Expression of *flh*, a mid/caudal diencephalon marker gene, increased at 6 hpf but returned to the control levels at 10 hpf in the *hif-3α∆42* mutant embryos (*Figure 6E–F*). Meanwhile, no difference in *pax2a* expression was observed between the mutant and the wild type embryos at 6 and 10 hpf (*Figure 6G–H*).

The lack of major phenotypes in body patterning was surprising. Excessive Wnt/β-catenin activity has been linked to major phenotypic changes in zebrafish. For example, the *masterblind (mbl)* mutant fish had no eyes, reduced telencephalon, and expanded diencephalon (*Heisenberg et al., 2001*). Likewise, treatment of zebrafish embryos with Lithium and BIO, which increase β-catenin signaling by inhibiting GSK3β, results in the absence of eyes or reduced eyes (*Kim et al., 2002*; *Nishiya et al., 2014*). A recent study has shown that zebrafish can activate compensation mechanisms to buffer against deleterious genetic mutations (*Rossi et al., 2015*). For instance, while knockdown of egfl7 resulted in major phenotypic changes, no phenotype was found in the *egfl7-/-* genetic mutants (*Rossi et al., 2015*). To test this possibility, we designed a morpholino (MO) targeting the Hif-3α2 ATG site. Injection of the Hif-3α2 MO but not the control MO reduced the Hif-3α2 protein levels (*Figure 7A—figure supplement 1A*). Injection of Hif-3α2 MO did not change the levels of Hif-3α1 mRNA (*Figure 7—figure supplement 1B*). Because the full-length Hif-3α1 protein is rapidly degraded under normoxia (*Zhang et al., 2012*), it could not be detected nor affected by the Hif-3α2 MO. Topflash reporter assay and qRT-PCR measurement showed that knockdown of Hif-3α2 significantly increased Wnt/β-catenin activity at 9 hpf (*Figure 7B–D*). Therefore, the Wnt/β-catenin activity is elevated in the morphants. Many of the morphants had phenotypes resembling those of *mbl* mutant embryos (*Heisenberg et al., 2001*), including the absent or reduced eyes and reduced forebrain (*Figure 7E*). These phenotypes were unlikely caused by MO toxicity because the Hif-3α2 targeting MO did not increase embryo mortality nor did it increase p53 mRNA expression at the injected dose (3 ng) (*Figure 7—figure supplement 1C*). Furthermore, the *hif-3α∆42* embryos were significantly less sensitive to the Hif-3α2 MO (*Figure 7F*). qRT-PCR results showed that the expression of *boz*, a maternal Wnt signaling target gene (*Leung et al., 2003*), increased at 6 hpf but returned to the control levels at 9 and 24 hpf in the *hif-3α∆42* mutants (*Figure 8A*). No change was detected in *boz* expression in the morphants at any of these stages (*Figure 8D*). The *vent* and *vox* mRNA levels increased significantly at 6 and 9 hpf but these increases disappeared at 24 hpf in the *hif-3α∆42* mutant embryos (*Figure 8B–C*). In contrast, knockdown of Hif-3α2 resulted in elevated *vent* and *vox* mRNA expression at 6, 9, as well as 24 hpf (*Figure 8E–F*). A reciprocal decrease was observed with *tp63* expression; *tp63* is a direct transcriptional target of Bmp signaling in zebrafish (*Bakkers et al., 2002*) (*Figure 8G–H*). These results support the notion that the *hif-3α∆42* mutant embryos have activated certain compensatory mechanisms. Next, we subjected the *hif-3α∆42* mutant embryos to Lithium and BIO. We reasoned that the *hif-3α∆42* mutant embryos should be more sensitive to these Wnt activating agents even if they can survive by activating as-yet-unidentified compensatory mechanisms. Indeed, when treated with Lithium and BIO, a significantly greater portion of the *hif-3α∆42* embryos exhibited the no eye or small eye phenotype compared to the wild type controls (*Figure 8I–J*).

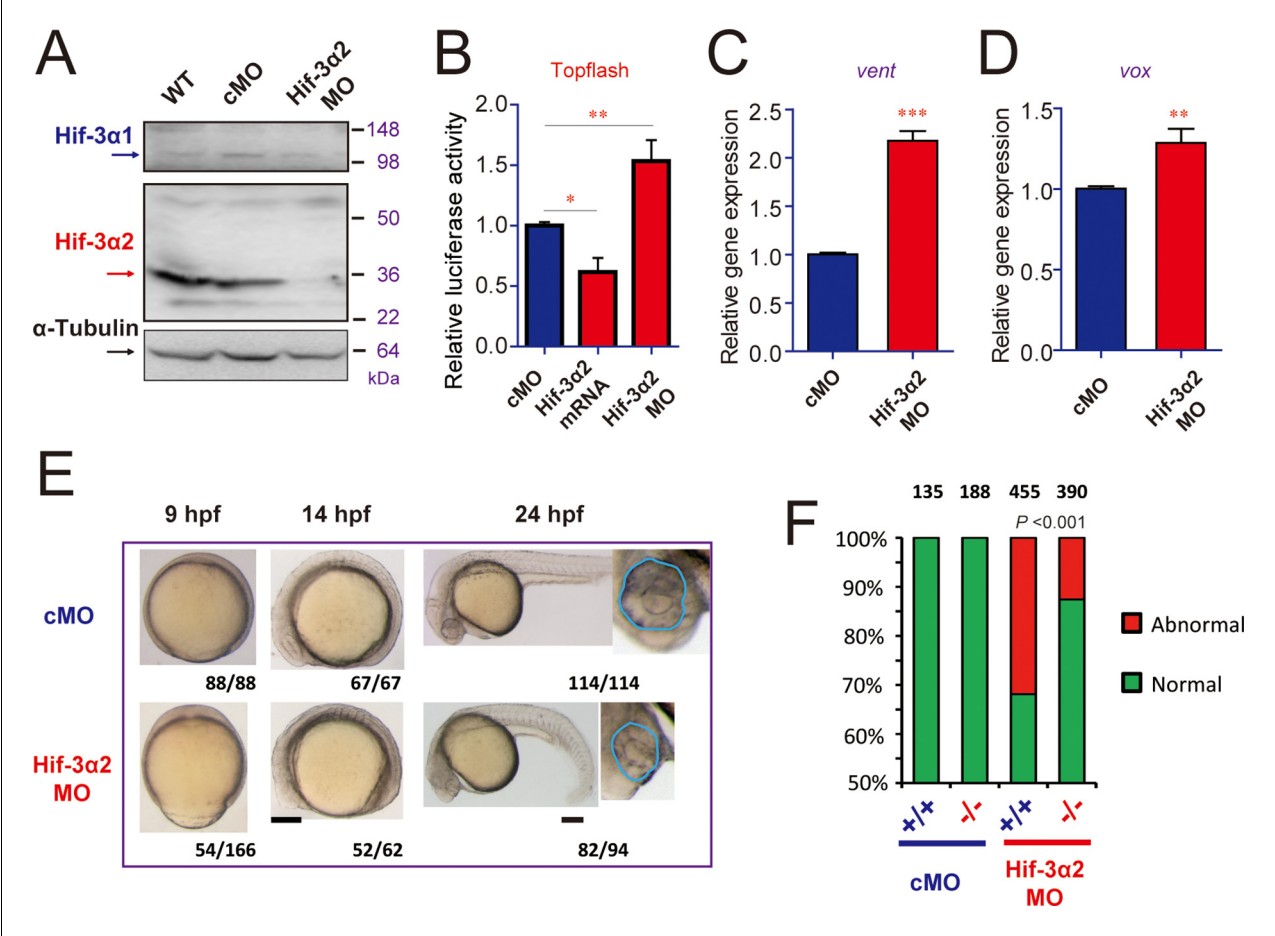

**Figure 7.** Hif-3α2 knockdown leads to Wnt/β-catenin signaling phenotypes. (A) Wild type (WT), control MO (cMO), and Hif-3α2 targeting MO (Hif-3α2 MO)-injected embryos were raised to 24 hpf in normoxic water. They were lysed and subjected to Western blotting. (B) Wild-type embryos injected with Topflash plasmid DNA together with the indicated capped mRNA or MO were raised to 9 hpf and the luciferase activity was measured. Values are means ± S.E. (n = 3). * and **p < 0.05 and 0.01. (C,D) Embryos injected with the indicated MOs were raised to 9 hpf and *vent* (C) and *vox* (D) mRNA levels were determined and presented as described above. (E) Phenotypes. The frequency of embryos with the indicated phenotypes is shown at the bottom of each panel. Scale bar = 200 μm. (F) *hif-3αΔ42* mutant embryos are less sensitive to Hif-3α2 MO. Control MO or Hif-3α2 MO were injected into wild type (+/+) or F3 homozygous *hif-3αΔ42* mutant embryos (-/-). The frequency of embryos with the abnormal phenotype (E) at 9 hpf was determined and shown. Total embryo number is shown on the top of each column. *P* value of Chi analysis is shown.

The following figure supplement is available for figure 7:

**Figure supplement 1.** Hif-3α2 targeting MO design and verification.

## Hif-3α2 binds to β-catenin and destabilizes the nuclear β-catenin complex

To investigate whether this action of Hif-3α2 is evolutionarily conserved and determine the underlying biochemical mechanisms, we tested the effects of Hif-3α2 in cultured HEK293T cells. Overexpression of Hif-3α2 abolished Wnt3a-induced Topflash reporter activity in these human cells (*Figure 9—figure supplemental 1A*). In contrast, Hif-3α2 had no effect on Wnt5a-induced non-canonical Ap1 reporter activity (*Figure 9—figure supplemental 1B*), suggesting that Hif-3α2 only inhibits the canonical Wnt signaling pathway. We engineered a truncated human HIF-3α9 by deleting its N-terminal 452 residues (*Figure 9—figure supplemental 1C*). This truncated HIF-3α9, like zebrafish Hif-3α2, had significant HRE-dependent transcriptional activity. It also inhibited Wnt3a- and β-catenin-induced Topflash activity (*Figure 9—figure supplemental 1D–E*). These data suggest that both activities are conserved in human HIF-3α.

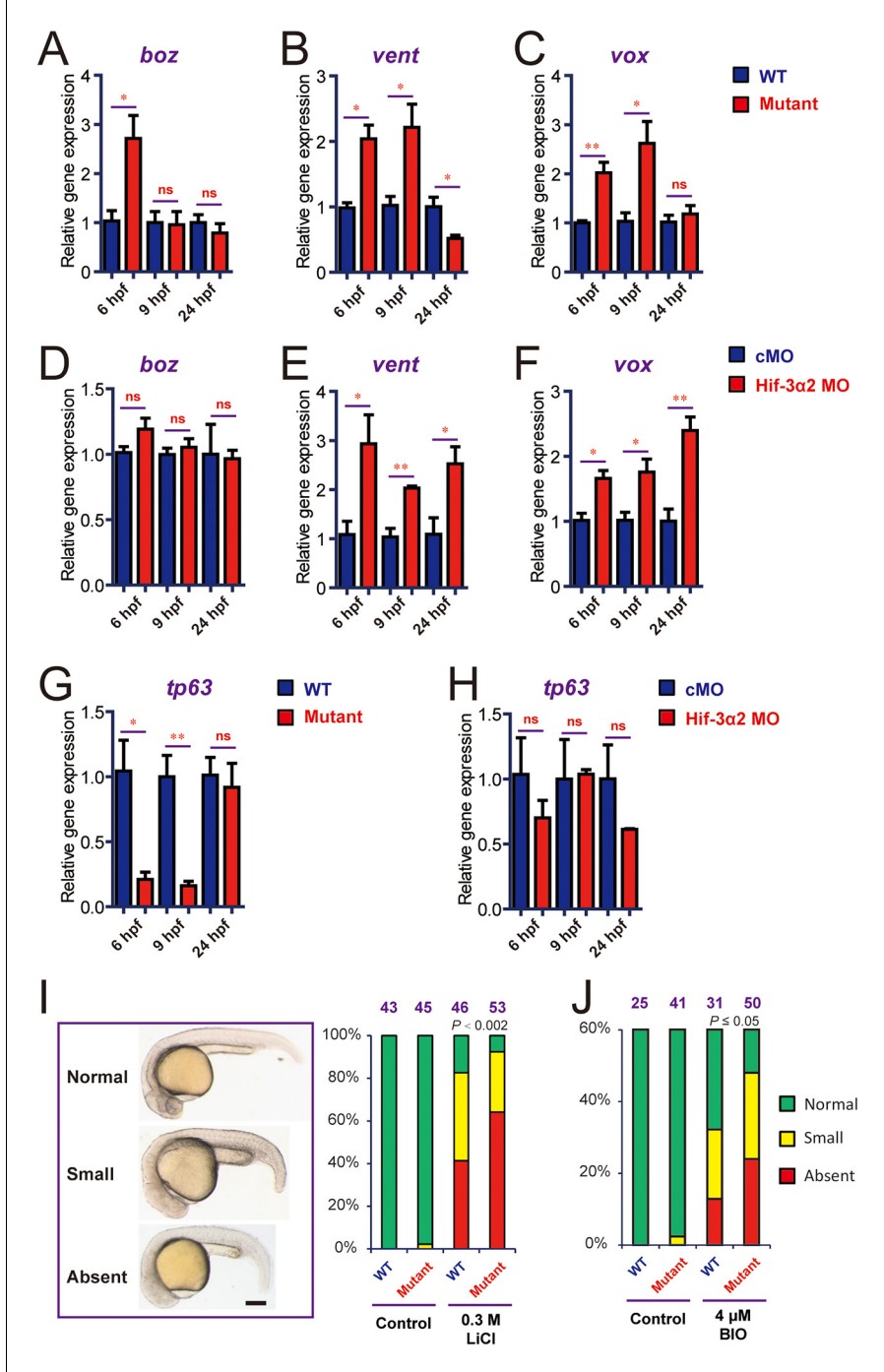

**Figure 8.** Genetic deletion but not knockdown of Hif-3α2 result in dynamic and compensatory changes in the expression of Wnt/β-catenin and BMP target genes. (A-C) Expression of *boz* (A), *vent* (B), and *vox* (C) in *hif-3αΔ42* mutant embryos at the indicated stages. (D-F) Expression of *boz* (D), *vent* (E), and *vox* (F) in morphants at the indicated stages. (G,H) Expression of tp63 in *hif-3αΔ42* mutant embryos (G) and morphants (H) at the indicated stages. In all above, the mRNA levels of the indicated genes were measure and presented as described above. Values are means +S.E. (n = 3). * and **p < 0.05 and P < 0.01. (I,J) Treatment of wild type and *hif-3αΔ42* mutant embryos with LiCl (0.3 mM) and BIO (4 µM) results in no eyes or small eyes phenotypes. Representative views are shown in the left panel and quantification results are shown in the right panel. The total embryo number is shown on the top. P value of Chi analysis is shown.

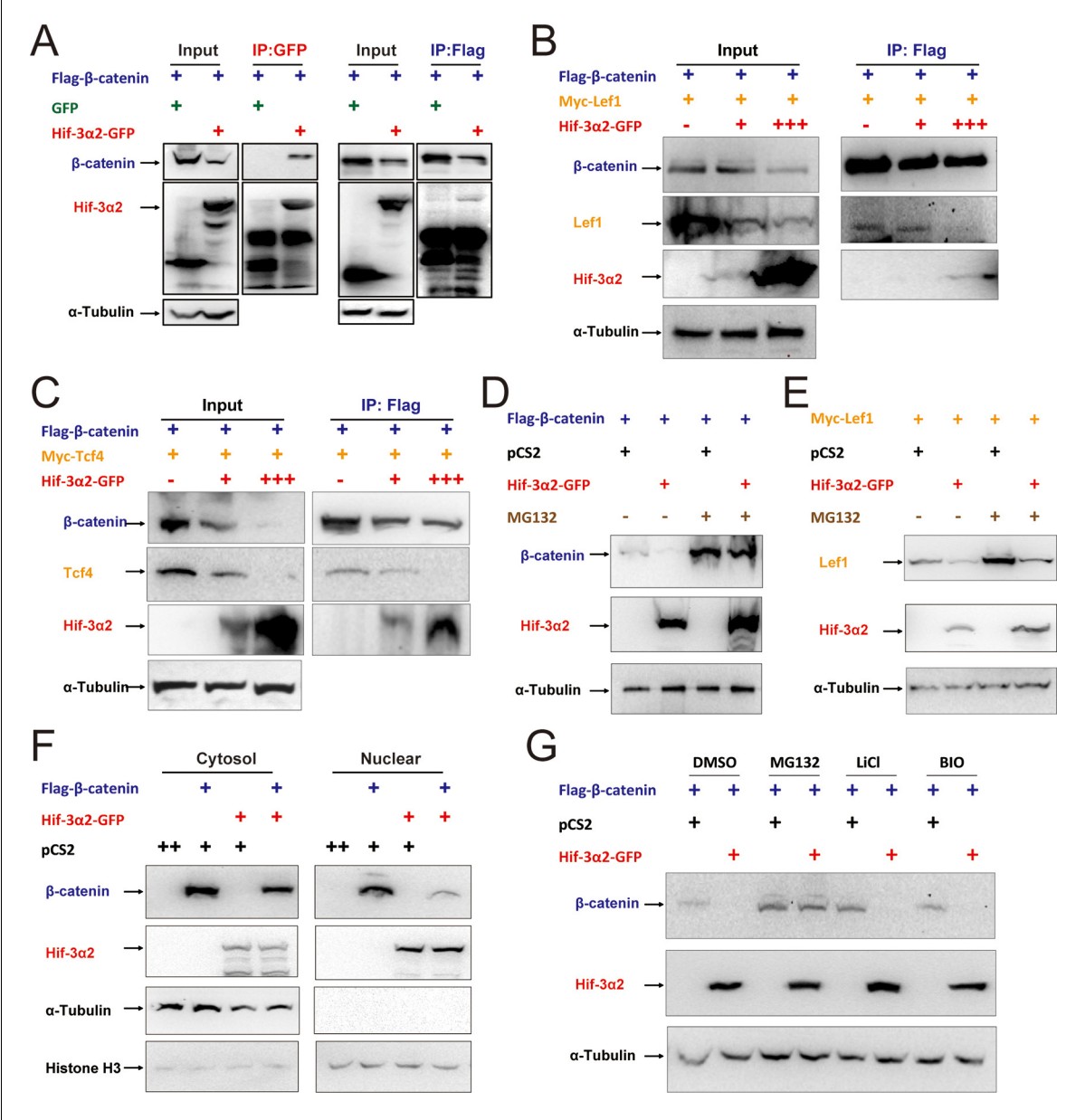

**Figure 9.** Hif-3α2 binds to β-catenin and destabilizes the nuclear β-catenin complex. (**A**) Co-IP experiment. HEK 293T cells were co-transfected with the indicated plasmids. 24 hr later, the cells were lysed and subjected to IP with an anti-GFP (left panel) or anti-Flag antibody (right panel) followed by immunoblotting using the indicated antibodies. (**B,C**) Dose-dependent effects of Hif-3α2 in reducing the β-catenin-Left1 complex (**B**) and β-catenin-Tcf4 complex (**C**). HEK 293T cells were co-transfected with the indicated plasmids. + and +++ indicate 1 and 3 μg DNA. 24 hr later, the cells were lysed and subjected to IP followed by Western blotting. (**D**) Hif-3α2 promotes β-catenin degradation. HEK 293T cells were co-transfected with the indicated plasmids. 24 hr after transfection, the cells were treated with MG132 (10 μM) for 12 hr and analyzed by Western blotting. (**E**) Hif-3α2 increases Lef1 degradation. HEK 293T cells transfected with the indicated plasmids. 24 hr after transfection, the cells were treated with MG132 (10 μM) for 12 hr and analyzed by Western blotting. (**F**) Hif-3α2 decreases nuclear β-catenin levels. HEK 293T cells transfected with the indicated plasmids were fractionated into nuclear and cytosolic fractions and analyzed by Western blotting. + and ++, indicate 1 and 2 μg DNA used. α-Tubulin and Histone H3 were measured as cytosolic and nuclear protein markers. (**G**) Lack of effects of GSKβ inhibitors. HEK293T cells were co-transfected with the indicated plasmids. 24 hr after transfection, the cells were treated with 6-bromoindirubin-3'-xime (BIO, 100 nM), LiCl (10 mM), or MG132 (10 μM) for 12 hr and lysed. The cell lysates were analyzed.

The following figure supplements are available for figure 9:

**Figure supplement 1.** The HER-dependent transcriptional activity and Wnt/β-catenin inhibitory activity are conserved in human cells.

*Figure 9 continued on next page*

*Figure 9 continued*

**Figure supplement 2.** Hif-3α2 binds to β-catenin but not Lef1 or Tcf4 directly.
**Figure supplement 3.** Hif-3α2 dose not affect Tcf3 stability.

Next, Flag-tagged β-catenin and GFP-tagged Hif-3α2 were co-transfected into HEK293T cells. Reciprocal co-IP experiments detected β-catenin and Hif-3α2 in the same complex (*Figure 9A*). A direct protein-protein interaction between β-catenin and Hif-3α2 was confirmed by GST pull-down assays (*Figure 9—figure supplement 2A*). No direct interaction was detected between Lef1 and Tcf4 and GST-Hif-3α2 (*Figure 9—figure supplement 2B–C*). To test whether the Hif-3α2 can compete for β-catenin binding with its nuclear partners, Flag-β-catenin and Myc-Lef1 were co-expressed with increasing levels of Hif-3α2-GFP. Co-expression of Hif-3α2-GFP decreased the levels of both β-catenin and Lef1 in the complex (*Figure 9B*). Likewise, overexpression of Hif-3α2-GFP also reduced the β-catenin and Tcf4 levels in the IP complex (*Figure 9C*). Analysis of the Western blotting results of the cell lysates showed that the levels of total β-catenin, Lef1, and Tcf4 were all decreased by Hif-3α2 in a dose-dependent manner (*Figure 9B–C*). Overexpression of Hif-3α2 had no effect on the levels of Tcf3 levels and Hif-3α2-GST failed to pull down Tcf3 (*Figure 9—figure supplement 3A–B*). We postulated that Hif-3α2 inhibits β-catenin by destabilizing β-catenin and its nuclear co-activators. To test this idea, the proteasome inhibitor MG132 was added. While Hif-3α2 expression resulted in a marked reduction in β-catenin levels, this reduction was blocked by MG132 treatment (*Figure 9D*). Likewise, MG132 treatment also partially inhibited the Hif-3α2-induced reduction in Lef1 levels (*Figure 9E*). MG132 treatment increased Hif-3α2 levels (*Figure 9D–E*). Next, HEK293T cells co-transfected with β-catenin and Hif-3α2-GFP were fractionated into nuclear and cytoplasmic fractions and analyzed. The expression of Hif-3α2 greatly reduced the levels of nuclear β-catenin, while it had a minimal effect on the levels of cytoplasmic β-catenin (*Figure 9F*). Addition of BIO and Lithium, two chemical inhibitors of GSKβ, did not affect the Hif-3α2-induced β-catenin degradation (*Figure 9G*). Taken together, these data indicate that Hif-3α2 binds to β-catenin and destabilizes the active β-catenin complex through a mechanism that is independent of GSKβ/β-TrCP-mediated degradation.

## Hif-3α2 regulates LR asymmetry by destabilizing β-catenin and this action is independent of its HRE-dependent transcriptional activity

Next, the molecular mechanisms underlying Hif-3α2's actions were investigated. While deletion of the TAD completely abolished HRE-dependent transcriptional activity (see *Figure 2D*), it only partially reduced the β-catenin inhibiting activity (*Figure 10A*). The ΔLZIP mutant had greater transcriptional activity (see *Figure 2D*) and full β-catenin inhibitory activity (*Figure 10A*). Next, several mutants were engineered to target several motifs conserved in the HIF/Hif-1α and -3α TAD (*Figure 10—figure supplement 1*). Changing L30D31L32 into A (i.e., M1 mutant) did not affect the β-catenin inhibitory activity but reduced the HRE-dependent transcriptional activity by half (*Figure 10A–B*). Mutant M2 (changing D44F45Q46 into A) had partial β-catenin inhibitory activity (*Figure 10A*), while it retained half of its HRE-dependent transcriptional activity (*Figure 10B*). The A36G mutant had full β-catenin inhibitory activity (*Figure 10A*), but completely lost its transcriptional activity (*Figure 10B*). Changing P37 to A, however, eliminated both activities (*Figure 10A, B*). The expression levels of these proteins were comparable (*Figure 10C*). These data suggest that Hif-3α2 inhibits.

The effects of these Hif-3α2 mutants in promoting β-catenin degradation were studied next. All three mutants possessing full β-catenin inhibitory activity, i.e., ΔLZIP, M1, and A36G, strongly induced β-catenin degradation (*Figure 10C*). P37A, which had no *Figure 10C*). M2 and ΔTAD, which had partial β-catenin inhibitory activity, had a modest impact on β-catenin stability (*Figure 10C*). Human HIF-3α9Δ 1–452 had similar activity in destabilizing β-catenin (*Figure 10—figure supplement 2*). Finally, the effects of the A36G and P37A mutants on the LR asymmetry development were determined. Overexpression of A36G and Hif-3α2 resulted in heart randomization and abnormal somite development (*Figure 10D, E*). In contrast, overexpression of P37A, which lacks the ability to induce β-catenin degradation, had little effect (*Figure 10D–E*). These findings suggest that Hif-3α2

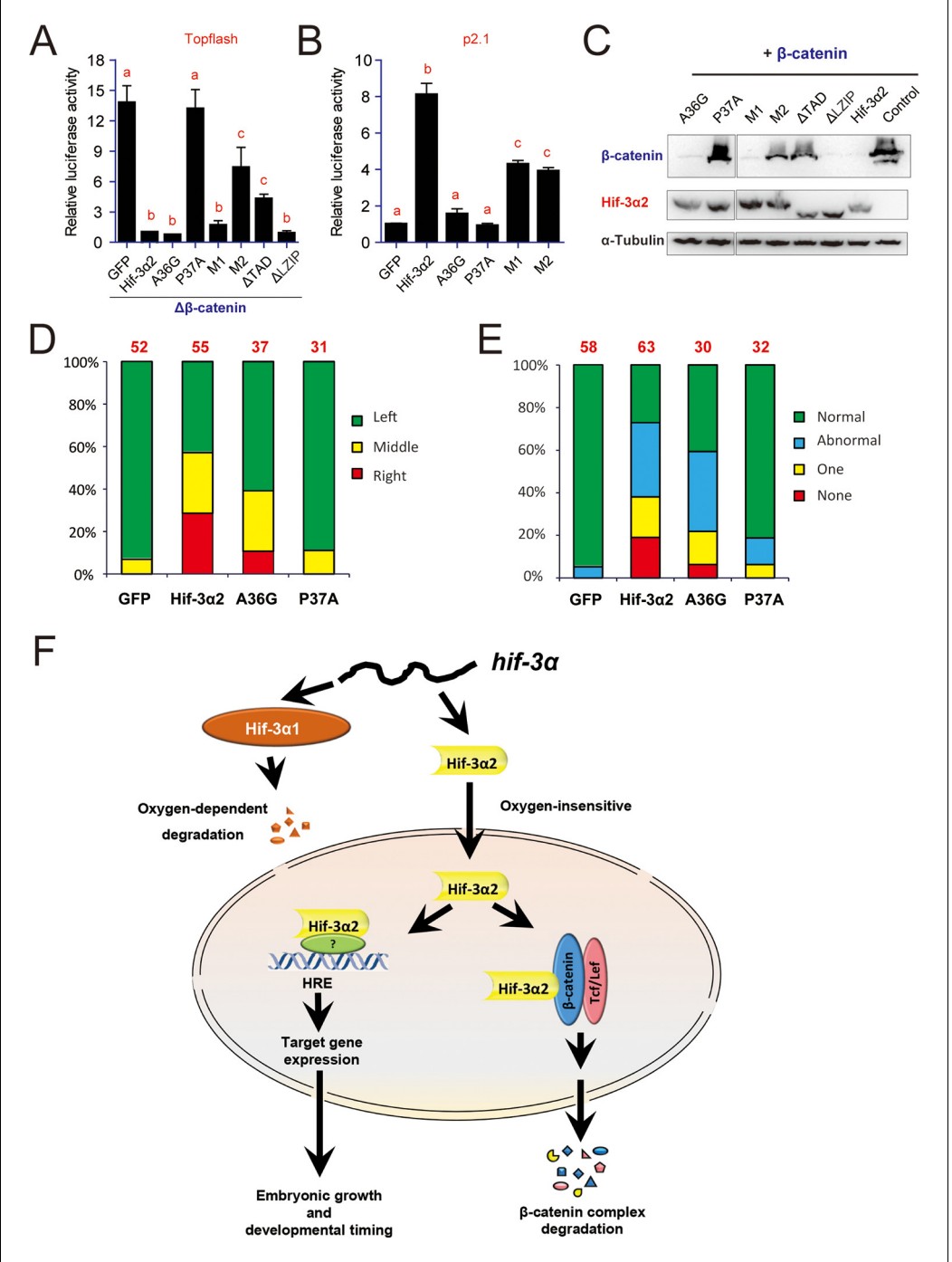

**Figure 10.** Hif-3α2 regulates LR asymmetry by destabilizing β-catenin independently of its transcriptional activity. (**A**) Inhibition of β-catenin-induced Topflash reporter activity. Zebrafish embryos were injected with 90 pg Topflash plasmid DNA together with capped mRNA (600 pg) of the indicated Hif-3α2 mutants (*Figure 10—figure supplement 1* for mutant details). Luciferase activity was measured at 9 hpf. Values are means $\pm$S.E. (n = 3). Groups labeled with different letters are significantly different from each other ($P < 0.05$). (**B**) HRE-dependent transcriptional activities of the indicated Hif-3α2 mutants. HEK293 cells were transfected with the indicated plasmid together with p2.1. The results are normalized and expressed as fold change over the GFP group and presented as described above. (**C**) Effects of Hif-3α2 mutants on β-catenin stability. HEK 293T cells were co-transfected with Flag-tagged β-catenin and the indicated Hif-3α2 mutants. The cell lysates were analyzed by Western blotting. (**D,E**) Effects of the indicated Hif-3α2 mutants on heart looping (**D**) and somite development (**E**). Embryos injected with the indicated capped mRNA were analyzed and scored as described in *Figure 3*. (**F**) *Figure 10 continued on next page*

*Figure 10 continued*

Oxygen-dependent and -independent action of Hif-3α. The *hif-3α* gene gives rise to Hif-3α1 (the full-length protein) and Hif-3α2 (a short spliced variant). Hif-3α1 is an oxygen-dependent transcription factor. It is rapidly degraded under normoxia. Under hypoxia, Hif-3α1 is stabilized and regulates the expression of many target genes, which in turn slow down growth and developmental timing (*Zhang et al., 2014*). Hif-3α2 is insensitive to oxygen tension. Hif-3α2 regulates target gene expression in an HRE-dependent manner under both normoxia and hypoxia. It is unclear whether Hif-3α2 can form a dimer with HIFβ. Hif-3α2 also binds to β-catenin and destabilizes the nuclear β-catenin complex. This Wnt signaling inhibitory action is independent of its HRE-dependent transcriptional activity.

The following figure supplements are available for figure 10:

**Figure supplement 1.** Sequence comparison of the TAD in the indicated human and zebrafish HIF/Hif proteins.

**Figure supplement 2.** Effects of full-length human HIF-3α9, its N-terminal deletion mutant, zebrafish Hif-3α1, and Hif-3α2 on β-catenin stability.

---

alters LR asymmetryby destabilizing β-catenin and this action is independent of its HRE-dependent transcriptional activity (*Figure 10F*).

## Discussion

In this study, we have identified a novel and oxygen-insensitive Hif-3α variant, Hif-3α2. To date, a number of HIF-3α protein-coding variants have been reported, including: a) the full-length protein; b) a HIF-3α variant similar to the full-length protein; c) HIF-3α variants lacking the LZIP domain; d) HIF-3α variants lacking both bHLH and LZIP domains; e) Short HIF-3α variants lacking the C-terminal half region; f) Truncated HIF-3α variants lacking the bHLH, ODD/N-TAD, and LZIP domains; and g) HIF-3α variants lacking the bHLH domain (*Duan, 2015*). Zebrafish Hif-3α2 has a structure distinct from those reported previously. Hif-3α2 lacks the entire N-terminal region and only contains the N-TAD and LZIP domains. Hif-3α2 is expressed in all embryonic stages and in many adult tissues examined. Although the Hif-3α2 mRNA levels were lower compared to those of the full-length Hif-3α1, the Hif-3α2 protein was abundantly expressed in embryonic and adult tissues under both normoxia and hypoxia. Functional analysis results show that Hif-3α2 has significant HRE-dependent transcriptional activity. This activity depends on its TAD. Although Hif-3α2 has the ability to induce HRE-dependent gene expression, its action is not identical to that of the full-length Hif-3α1. The transcriptional activity of Hif-3α2 is weaker compared to that of Hif-3α1. Among the five Hif-31α target genes examined, Hif-3α2 increases the expression of *igfbp-1a, mlp3c* and *redd1* but not that of *zp3v2* and *sqrdl*.

We observed two major phenotypes resulting from Hif-3α2 overexpression. One is the global growth retardation and developmental timing delay. This is similar to what has been reported for Hif-3α1 (*Zhang et al., 2014*). This action apparently requires the transcriptional activity because it was abolished by deletion the TAD but not LZIP domain. Another phenotype is the defective LR asymmetry. Forced expression of Hif-3α2 altered the expression of several asymmetry genes and resulted in the randomization of laterality of the heart, liver, and pancreas. This action is unique to Hif-3α2 because no such phenotype was observed in the Hif-3α1 expressing embryos. Importantly, the A36G mutant, which had no HRE-dependent transcriptional activity, caused a similar heart randomization, suggesting that this action does not required its HRE transcriptional activity. The LR axis development in zebrafish is divided into several phases, beginning from the symmetry breaking, and continuing with DFC appearance and clustering, KV organogenesis, lateral plate mesoderm asymmetry, and left- or right-specific organ formation (*Carl et al., 2007*; *Matsui and Bessho, 2012*). Our results suggest that forced expression of Hif-3α2 impairs LR axis development at the step of KV organogenesis/function. This conclusion is supported by the findings that Hif-3α2 overexpression altered KV morphology and *spaw* expression, while it did not affect DFC appearance or DFC clustering.

We provided several lines of evidence suggesting that Hif-3α2 impairs KV development by inhibiting the canonical Wnt/β-catenin signaling pathway. These include: i) Hif-3α2 overexpression

decreases Wnt target gene expression while it had no effect on Fgf-, Hedgehog-, and Notch target genes; ii) Co-expression of Hif-3α2 inhibits Wnt3 and β-catenin-induced gene expression and dorsalization; iii) genetic deletion of Hif-3α2 (without affecting Hif-3α1 expression and function) increases Topflash reporter activity and Wnt target gene expression; iv) MO-based knockdown had similar effects in elevating Wnt signaling. These data strongly suggest that Hif-3α2 is a negative regulator of canonical Wnt signaling. Wnt signaling plays multiple roles in vertebrate axis specification (see *Hikasa and Sokol, 2013*). While the maternal Wnt/β-catenin signaling contributes to the dorsoventral axis establishment in early embryogenesis, the zygotic Wnt/β-catenin signaling is involved in anteroposterior axis specification. It has been suggested that these two different roles are mediated by two distinct sets of target genes (*Hikasa and Sokol, 2013*). As discussed earlier, canonical Wnt/β-catenin signaling also plays an important role in KV development and the establishment of the LR axis in zebrafish. Our results showed that overexpression of Hif-3α2 acts at the step of KV development. This may be related to the fact that it takes several hours for the injected mRNA to be translated into protein, and for resulting biological activities to manifest. Since Hif-3α2 mRNA is maternally deposited, the endogenous Hif-3α2 may act earlier. Indeed, we observed a significant increase in maternal Wnt signaling and elevated mortality in early stages in the mutant embryos. Although deletion of Hif-3α2 by either genetic mutations or gene knockdown results in elevated Wnt/β-catenin signaling, the two approaches gave rise to very different phenotypes. The F3 *hif-3αΔ42* mutant embryos had significantly elevated mortality in early stages. This increase in embryo mortality was also observed in F3 *hif-3αΔ20* mutant embryos. In contrast, no mortality increase was detected in the morphants. In both genetic alleles, the vast majority of the surviving embryos looked morphologically normal. The morphants, however, displayed phenotypes characteristic of excessive Wnt signaling. There are several possible explanations for the different phenotype. It is conceivable that the lack of a body patterning phenotype in the surviving mutant embryos may be due to activation of compensatory mechanisms. Such compensation mechanisms have been demonstrated recently in zebrafish and accounted for the different phenotypes found in the *egfl7-/-* genetic mutants and Egfl7 morphants (*Rossi et al., 2015*). In support of this view, the *hif-3αΔ42* mutant embryos displayed a transient and dynamic activation of Wnt signaling, while a sustained elevation in Wnt signaling was observed in the morphants. A concomitant reduction in BMP signaling was observed in the *hif-3αΔ42* embryos but not in the morphants. Furthermore, the *hif-3αΔ42* mutant embryos were more sensitive to the Wnt activating agents Lithium and BIO. Another possible explanation is the differential effects of genetic mutations and MO knockdown on the maternal Wnt signaling. The maternal Wnt signaling is significantly elevated in the F3 *hif-3αΔ42* mutant embryos, as indicated by the increased boz expression, while no such increase was observed in the morphants. It is also possible that Hif-3α2 may have other activities in vivo. In fact, our results show that Hif-3α2 can regulate gene expression via its TAD domain. We cannot exclude the possibility Hif-3α2 may affect other pathways directly or indirectly.

A biochemical and functional interaction between an HIF-3α isoform and Wnt signaling has not been reported before. However, HIF-1α has been shown to both inhibit and activate Wnt signaling in mammals. Kaidi et al. (*Kaidi et al., 2007*) reported that hypoxia inhibits the β-catenin/TCF4 complex formation and activity, resulting in a G1 arrest in cultured colon cancer cells. These authors proposed that HIF-1α acts by competing with TCF4 for β-catenin binding. Lim et al. (*Lim et al., 2008*) reported that HIF-1α inhibits Wnt signaling by binding to human arrest defective 1 (hARD1), which is responsible for β-catenin acetylation. Another mechanism was reported by *Chen et al. (2013)*. They showed that HIF-1α indirectly inhibits the Wnt signaling pathway by inducing the expression of a Wnt antagonist Sclerostin in MC3T3 osteoblastic cells. In a very recent paper, Majmundar et al. demonstrated that tissue-specific knockout of HIF-1α in mice has little effect on embryonic and fetal myogenesis but increases adult muscle regeneration (*Majmundar et al., 2015*). HIF-1α acts by repressing canonical Wnt signaling in adult skeletal muscle. HIF-1α has also been shown to activate Wnt signaling in mammals. Using β-Gal Wnt reporter and Hif-1a conditional knockout mouse lines, *Mazumdar et al. (2010)*Mazumdar et al. (2010) showed that HIF-1α increases Wnt/β-catenin signaling activity in murine embryonic mesencephalon and adult hippocampal neuronal stem cells and precursor cells in vitro and in vivo. These authors further showed that HIF-1α activates canonical Wnt signaling by increasing the expression levels of β-catenin and its downstream effectors LEF-1 and TCF-1. Subsequently, Jeong and Pack reported that HIF-1α increases β-catenin expression level and action in SH-SY5Y human neuroblastoma cells (*Jeong and Park, 2013*). Medley et al. reported that

hypoxia activates Wnt signaling in undifferentiated iPS cells (*Medley et al., 2013*). As in the case of HIF-1α, HIF-2α has been shown to inhibit or/and enhance Wnt/β-catenin signaling (*Choi et al., 2010*; *Santoyo-Ramos et al., 2014*). Another mechanism by which HIF-1α and -2α can active the Wnt signaling pathway was reported by Yuen et al. recently (*Yuen et al., 2014*). They reported that HIF-1α and -2α directly bind to the promoters of the Wnt7a and 7b genes and induce their expression in murine oligodendrocytes. Our study is different from the above studies in that Hif-3α2 is an oxygen-insensitive and constitutively expressed Hif-3α isoform and therefore functions under both normoxia and hypoxia. Importantly, our biochemical analysis results revealed that Hif-3α2 directly binds to β-catenin and promotes the degradation of the nuclear β-catenin complex. This action is specific, as shown by the fact that Hif-3α2 overexpression had no effect on the level of Tcf3, a β-catenin suppressor. This mechanism is distinct from those reported for HIF-1α and -2α.

In addition to the GSK3β/β-TrCP-mediated degradation mentioned earlier, β-catenin is also subject to negative regulation by pVHL. The β-catenin degradation induced by pVHL is dependent on the presence of Jade-1, an E3 ubiquitin ligase (*Chitalia et al., 2008*). These two pathways are unlikely to be involved in the Hif-3α2-induced degradation for several reasons. First, both β-TrCP and Jade-1 bind to the N-terminal region of β-catenin in a phosphorylation-dependent manner and predominantly degrade β-catenin in the cytosol. Our results showed that Hif-3α2 inhibits the activity of β-cateninΔN, a constitutively active β-catenin mutant lacking the N-terminal sequence. Second, two GSK-3β inhibitors, BIO and lithium, did not inhibit Hif-3α2-induced β-catenin degradation. Finally, Hif-3α2 not only reduces the levels of β-catenin but also decreases the levels of Tcf-4 and Lef1. GST-Hif-3α2 did not directly interact with Tcf-4 and Lef1. Therefore, Hif-3α2 probably binds to β-catenin, which in turn interacts with Tcf-4 and Lef1. It is unclear whether other factors may be involved in the physical association between Hif-3α2 and β-catenin and whether post-translational modification of Hif-3α2 and/or β-catenin is involved its their binding. β-catenin can shuttle in and out of the nucleus (*Krieghoff et al., 2006*). The results of our cell fractionation experiment showed that Hif-3α2 expression resulted in a marked decrease in the nuclear β-catenin levels without notable changes in the cytosolic β-catenin levels. In general, the fate of active β-catenin in the nucleus is not well understood. A recent study reported that the RING finger E3 ubiquitin ligase c-Cbl preferentially targets the active nuclear β-catenin for degradation by binding to the ARM domains of β-catenin (*Chitalia et al., 2013*). Wnt activation promotes c-Cbl phosphorylation at Y371, and the phosphorylated c-Cbl dimerizes, translocates into the nucleus and promotes nuclear active β-catenin degradation during the Wnt-on phase (*Shivanna et al., 2015*). TRIM33, another E3 ubiquitin ligase, has been reported to interact with and ubiquitylate nuclear β-catenin, and promote nuclear β-catenin degradation in a GSK-3β and β-TrCP-independent manner (*Xue et al., 2015*). It is unknown where c-Cbl and/or TRIM33 mediated the degradation of the β-catenin/Tfc4/Left1 complex. Future studies will be needed to determine the molecular mechanism(s) responsible for the Hif-3α2-induced β-catenin complex destabilization.

Another intriguing and potentially important finding made in this study is that Hif-3α2 has HRE-dependent transcriptional activity. The current dogma is that HIFs act as α/β heterodimers (*Semenza, 2012*). The bHLH domain in HIFα is known to be involved in DNA binding, and the PAS-A and PAS-B domains are proposed to be involved in dimerization with HIF-β and target gene specificity (*Erbel et al., 2003*; *Simon and Keith, 2008*). Although Hif-3α2 has a TAD, it lacks the bHLH and PAS domains. While our findings suggest that Hif-3α2 has transcriptional activity under normoxia, it is not clear whether this action requires HIF-β. *Makino et al. (2001)* reported that mouse IPAS could not form dimers with HIF-β. Rather, it formed a complex with HIF-1α but this complex did bind to an HRE. Using co-immunoprecipitation assays, *Maynard et al. (2005)* showed that HIF-3α4 interacted with HIF-1α when transfected into HEK293 cells. HIF-3α4 also formed a complex with HIF-β. The HIF-3α4/HIF-β complex did not appear to be capable of binding to an HRE, but it inhibited the binding of the HIF-1α/β complex to the HRE in a dose-dependent fashion. A similar relationship between HIF-3α4 and HIF-2α was reported (*Maynard et al., 2007*). The structure of mouse IPAS and human HIF-3α4 is very different from zebrafish Hif-3α2. Mouse IPAS and human HIF-3α4 contain bHLH, PAS-A and -B, and PAC domains but lack the ODD/T-NAD and LIZP domains. In contract, zebrafish Hif-3α2 contains N-TAD and LZIP only. The LZIP domain is only found in HIF-3α among the 3 HIFα isoforms (*Duan, 2015*). LZIP is the dimerization domain of the ATF-6/CREB subfamily of bZIPs (a class of eukaryotic transcription factors) (*Vinson et al., 1989*) and it facilitates dimerization and in some cases higher oligomerization of proteins (*Lu et al., 1997*). LZIP containing

regulatory proteins include c-Fos and c-Jun and the LZIP domains in Fos and Jun are necessary for the formation of the AP1 heterodimer (*Ransone et al., 1989*). The LZIP domains in Fos and Jun have also been shown to bind to cAMP-responsive elements as a homodimer and can activate transcription from CRE-containing reporter genes (*Lu et al., 1997*; *Lu et al., 1998*). Our available data showed that the ΔLZIP mutant had stronger transcriptional activity. Future studies are needed to elucidate whether Hif-3α2 exerts its HRE-dependent activity independently of Hif β and to determine the functional relationships between Hif-3α2, Hif-3α1, and other Hifα proteins.

In summary, Hif-3α2 represents a novel and $O_2$-insensitive HIF/Hif-3α variants. Hif-3α2 has two distinct functions: activating HRE-dependent gene expression and binding to β-catenin and destabilizing the nuclear β-catenin complex. The corresponding region of human HIF-3α possesses both activities, suggesting that both actions are evolutionarily conserved. In agreement with this notion, Wnt3a knockout mice exhibit randomized internal organ positioning across the midline (*Nakaya et al., 2005*). The Hif-3α$^{-/-}$ mice had enlarged embryonic right atrium and right ventricle and this phenotype became more prominent in neonatal mice and continues until the adult stage (*Yamashita et al., 2008*). In contrast, no enlargement was observed in the left ventricle in these mutant mice (*Yamashita et al., 2008*). Our mutation analysis results suggested that different structural element(s) are responsible for these two distinct biological actions. Interestingly, the full-length Hif-3α1, while exhibiting strong HRE-dependent transcriptional activity, does not inhibit β-catenin activity. It is possible that there are structural element(s) in the full-length Hif-3α proteins that mask this important activity. In support of this view, we found that while the full-length human HIF-3α9 has no β-catenin inhibitory activity, deletion of its N-terminal sequence can unmask this ability (*Figure 10—figure supplement 2*). An earlier study suggested that there are structural element(s) in the full-length human HIF-3α inhibiting its ubiquitylation (*Maynard et al., 2003*). Sequence analysis of human and fish HIF/Hifαs indicate that the residues critical for promoting β-catenin degradation are conserved in human and fish HIF/Hifαs. It is possible that all HIFαs may have the ability to interact with β-catenin. Future studies will determine whether this activity is conserved in the full-length HIFαs and reveal how they are regulated. This would explain the inconsistent and even opposite results regarding the interactions between HIF-1/2α and Wnt/β-catenin signaling.

## Materials and methods

### Experimental procedures

#### Reagents

All chemicals were purchased from Fisher Scientific unless otherwise noted. PCR primers, superscript III reverse transcriptase, restriction enzymes, cell culture media, antibiotics, fetal bovine serum, and trypsin were purchased from Invitrogen. MG132, 6-bromoindirubin-3'-xime (BIO), lithium chloride, and anti-Flag antibody were purchased from Sigma-Aldrich. T7 Endonuclease I and Phusion High-Fidelity DNA Polymerase were obtained from New England Biolabs. Antibodies against GFP, Myc, phospho-Akt, phospho-Erk, total Akt, total Erk, Histone H3, and α-Tubulin were purchased from Torrey Pines Biolabs, Clontech, Cell Signaling Technology, Santa Cruz Biotechnology, and Sigma, respectively. Goat anti-mouse and rabbit light chain specific secondary antibodies were bought from Jackson Immuno Research. The zebrafish β-catenin plasmid was a gift from Dr. Anming Meng, Tsinghua University. The Myc-lef1 and myc-Tcf3 plasmids were kindly provided by Dr. Richard Dorsky, University of Utah, and *charon*, *spaw*, and *lft2* plasmids were provided by Dr. Joseph Yost, University of Utah, The Myc-Tcf4 plasmid was a gift from Wei Wu, Tsinghua University. The pcDNA-Zeo(-)-HIF-3a-1 plasmid was kindly provided by Dr. Johanna Myllyharju, University of Oulu. The zebrafish codon-optimized Cas9 construct was a gift from Dr. Wenbiao Chen, Vanderbilt University School of Medicine.

#### Animals

Wild type and mutant zebrafish (*Danio rerio*) were maintained as previously reported (*Zhang et al., 2014*). The LiPan transgenic fish were obtained from Dr. Wenbiao Chen, Vanderbilt University School of Medicine. Animal handling was conducted following guidelines approved by the University of Michigan Committee on the Use and Care of Animals.

## Cloning and plasmid construction

The Hif-3α2 full-length cDNA sequence was obtained by 5'- and 3'- RACE following published methods (*Funkenstein et al., 2002*). Genomic structure was determined by comparing the full-length Hif-3α1 and -3α2 cDNA sequences and the zebrafish genome sequence (http://www.ensembl.org/Danio_rerio/index.html, http://genome.ucsc.edu/cgi-bin/hgBlat). The amino acid sequence alignment was performed using the DNASTAR MegAlign program ClustalW Method. Hif-3α2open reading frame (ORF) was amplified by PCR using F_ and R_full-length Hif-3α2 primers, cloned into the pBluescript SK(-) vector, and sequenced. For functional analysis, Hif-3α2ORF DNA was subcloned into pCS2-eGFP and pCS2-Flag vectors. To determine the Hif-3α2domains and motifs important for its biological functions, several truncation and point mutants were engineered. The ΔTAD mutant was generated by deletion of the N-terminal 55 aa. The ΔLIZP mutant was made by deleting the C-terminal 45 aa. Mutants A36G, P37A, L47S, M1 (L30D31L32→AAA), and M2 (D44F45Q46→AAA) were generated by site-directed mutagenesis using Stratagene's QuikChange II mutagenesisKit (Agilent Technologies). For GST pull down assay, Hif-3α2 was cloned into the pGEX-KG vector to create Hif-3α2-GST fusion protein construct. The primers used for constructing these plasmids are shown in *Supplemental file 1*. All plasmids were verified by DNA sequencing. The construction of the stabilized Hif-1α (Hif-1α'), Hif-3α1 (Hif-3α1'), and β-cateninΔN was previously reported (*Rong et al., 2014*; *Zhang et al., 2012*; *Zhang et al., 2014*).

## Hypoxia and chemical treatment

Zebrafish embryo hypoxia treatment was conducted as described previously (*Kajimura et al., 2005*). For LiCl treatment, 8 hpf wild type or mutant embryos were exposed to 0.3 mM LiCl for 15 min, washed with embryo medium, and grew to 24 hpf. For BIO treatment, 6 hpf wild type or mutant embryos were exposed to 4 μM BIO for 18 hr, washed with embryo medium, and photographed. The cultured cells were treated with 10 mM LiCl, or 100 nM BIO, or 10 uM MG132 for 12 hr.

## Microinjection

Capped mRNAs were synthesized using the mMESSAGE mMACHINE SP6 kit (Ambion). Linearized plasmids were used as templates. A Hif-3α2 targeting MO (5'- ACTTCTCCACTCCTTCCATCTCCAT-3') and a standard control MO (5'- CCTCTTACCTCAGTTACAATTTATA-3') were purchased from Gene Tools. Capped mRNA or MO was microinjected into zebrafish embryos at the 1- to2-cell stage following previously reported procedures (*Zhou et al., 2008*).

## Cell culture

Human HEK293T, HEK293, Hela, and U2OS cells were purchased from American Type Culture Collection. Cell culture, transfection, subcellular location, and luciferase assays were performed as described previously (*Zhang et al., 2012*).

## Subcellular localization and luciferase reporter assay

To determine the subcellular localization and transcription activity of Hif-3α2 and its mutants, embryos were injected with the corresponding capped mRNA and raised to 6 hpf (hours post fertilization) under normoxia. The GFP signal was observed under fluorescence microscopy. Luciferase reporter assay was carried out using Dual-Luciferase Reporter Assay System (Promega) as previously reported (*Zhang et al., 2012*).

## RT-PCR, qPCR, and whole-mount in situ hybridization

Total RNA extraction and reverse transcription were carried out as previously reported (*Zhang et al., 2014*). PCR and RT-PCR were performed using Taq DNA polymerase (New England Biolabs). PCR amplification was 2-min incubation at 95℃, followed by 32 cycles of 30 s at 94℃, 30 s at 56℃, and 30 s or 90 s at 68℃. PCR products were resolved by 1.5% agarose gel electrophoresis and visualized by ethidium bromide staining. Quantitative real-time RT-PCR (qRT-PCR) and analysis were done as described before (*Kamei et al., 2008*). Primers used are shown in *Supplemental file 1*. Whole-mount in situ hybridization was performed following a previously published procedure (*Maures and Duan, 2002*).

## Co-immunoprecipitation (IP) and GST pull-down assay

Transiently transfected HEK293T cells in 100-mm dishes were harvested by scraping directly into Lysis buffer (20 mM Tris-HCl pH 8, 137 mM NaCl, 1% Nonidet P-40, 2 mM EDTA, protease inhibitors), sonicated, and centrifuged. Cell lysates were pre-cleared by incubation with Protein-A/G beads (EMD Millipore) at 4°C for 30 min, then incubated with the desired antibody overnight at 4°C and immunoprecipitated by incubating with Protein-A/G beads at 4°C for 2 hr. The beads were washed with wash buffer (10 mM Tris pH7.4, 1 mM EDTA, 1 mM EGTA, 150 mM NaCl, 1% Triton X-100, 0.2 mM sodium orthovanadate, protease inhibitors) four times and boiled in 1X PBS. GST pull-down assay was performed using Glutathione Sepharose 4B (GE Healthcare) following a previously published protocol (*Xu et al., 2004*). Immunoprecipitates or total cell lysates were analyzed by Western immunoblotting as described below.

## Cell fractionation and Western blotting

To detect the effect of Hif-3α in promoting β-catenin degradation in different cellular compartments, HEK293T cells transfected with plasmids of interest were fractionated into nuclear and cytoplasmic proteins using the Nuclear/Cytosolic Fractionation Kit (Cell Biolabs) following the manufacturer's instruction. Protein concentrations were determined using the Thermo Scientific Pierce BCA Protein Assay Kit (Thermo Fisher Scientific). To detect the level of endogenous Hif-3α, a validated polyclonal antibody was used (*Zhang et al., 2012*). Fish were homogenized in lysis buffer and the transfected cells were lysed by adding lysis buffer. These samples were subjected to immunoblotting analysis as described previously (*Zhang et al., 2012*).

## CRISPR/Cas9 genome editing mediated deletion of Hif-3α2

CRISPR/Cas9-based genome editing experiments were carried out to genetically ablateHIF-3α2. The DNA template for Hif-3α2 gRNA (5'- GCCCCGCTGAAGAGCTGCCCA-3') synthesis was generated by PCR using the pDR274 plasmid (Addgene) as template and primers shown in *Supplemental file 1*. In vitro transcription was carried out using 200 ng template DNA and T7 RNA polymerase (Promega). Zebrafish codon-optimized Cas9 plasmid was linearized by XbaI digestion, and Cas9 capped mRNA was transcribed using a mMESSAGE mMACHINE T3 kit (Ambion). The size and quality of the capped mRNA and gRNA were confirmed by electrophoresis using a 2% (wt/vol) agarose gel. One-cell stage embryos were injected with 1 nl solution containing 200 ng/μl Cas9 mRNA and 20 ng/μl gRNA. The injected embryos were raised to adulthood. F0 fish were crossed to wild type fish to generate F1 progeny. F1 embryos were genotyped by T7E1 mutagenesis assay, NcoI digestion and DNA sequencing. Heterozygous F1 fish were crossed to generate F2 fish. The F2 fish were genotyped. The stable lines were maintained by inbred crossing. F3 homozygous embryos were used for phenotypic and molecular analysis.

## T7EI mutagenesis assay, *Nco*I mutagenesis assay, and sequencing

To detect indel, genomic DNA was isolated from pooled embryos or adult tail. A 480-bp DNA fragment was amplified by PCR using Phusion High-Fidelity DNA Polymerase (New England Biolabs) and the primers shown in *Supplemental file 1*. The genomic PCR product was purified using the QIAquick PCR purification kit (Qiagen). 200 ng purified DNA was mixed with NEBuffer 2 (New England Biolabs), denatured at 95°C for 5 min, and cooled down to 85°C at -2°C per second, then to 25°C at -0.1°C per second, and finally to 4°C for 5 min to form DNA heteroduplex. The annealed DNA was digested with 1 unit of T7 endonuclease I for 1 hr at 37°C or 10 units *Nco*I (Promega) at 37°C for 3 hr. The reaction was stopped by adding 1 μL 500 mM EDTA. The digested products were separated in a 2% agarose gel. After confirming del using the T7E1 and *Nco*I mutagenesis assays, the genomic DNA fragment was subcloned into pGEM-T (Promega) and sequenced. Primers used are shown in *Supplemental file 1*.

### Statistics

The data shown are means ± SE. Differences among groups were analyzed by one-way or two-way ANOVA or by Student's *t* test using GraphPad Prism. Significance was accepted at $P < 0.05$ or lower.

## Acknowledgements

We are grateful to Dr. Kenneth Cadigan, University of Michigan, and Ms. Chunyang Zhang, Indiana University, for reading and commenting on an early version of this manuscript. We thank Mr. John Allard for proofreading and editing this manuscript. This study was supported by NSF grant IOS-1051034 and MCube2.0 Project U0496246.

## Additional information

### Funding

| Funder | Grant reference number | Author |
| --- | --- | --- |
| National Science Foundation | IOS-1051034 | Cunming Duan |

The funders had no role in study design, data collection and interpretation, or the decision to submit the work for publication.

### Author contributions

PZ, Final approval of the version to be published, Conception and design, Acquisition of data, Analysis and interpretation of data; YB, Final approval of the version to be published, Acquisition of data, Analysis and interpretation of data; LL, YL, Final approval of the version to be published, Analysis and interpretation of data, Contributed unpublished essential data or reagents; CD, Conception and design, Analysis and interpretation of data, Drafting or revising the article

### Author ORCIDs

Cunming Duan, http://orcid.org/0000-0001-6794-2762

### Ethics

Animal experimentation: This study was performed in strict accordance with the recommendations in the Guide for the Care and Use of Laboratory Animals of the National Institutes of Health. All of the animals were handled according to approved institutional animal care and use committee (IACUC) protocols (#09707) of the University of Michigan.

## Additional files

### Supplementary files

• Supplemental file 1. PCR primers used in this study.

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
