## [Decision Letter]

[Editors’ note: this article was originally rejected after discussions between the reviewers, but the authors were invited to resubmit after an appeal against the decision.]

Thank you for choosing to send your work entitled "An oxygen-insensitive Hif-3α isoform regulates left-right asymmetry by destabilizing the nuclear β-catenin complex" for consideration at *eLife*. Your full submission has been evaluated by Charles Sawyers (Senior Editor), Tanya Whitfield (Reviewing Editor), and three peer reviewers. The decision was reached after discussions between the reviewers and the editors. Based on our discussions and the individual reviews below, we regret to inform you that your work will not be considered further for publication in *eLife*.

As you will see, all three reviewers found merit in your study and recognised that this is an important and complex field of study. However, at present there are too many concerns to make your manuscript suitable for publication in *eLife* without major revisions that would be too substantial for what we require at *eLife*. In particular, it was felt that explicit clarification of the effects of your mutation on the various HIF3 isoforms was needed, together with a much more rigorous analysis of the mutant phenotype, particularly in relation to Wnt signalling. The effects on left-right asymmetry, mentioned in the title, were not fully convincing and needed additional support. We would recommend that a more streamlined version of the manuscript, delivering a clearer key message, would be helpful.

Reviewer #1:

Zhang and colleagues analyse the function of a novel hypoxia-inducible factor 3a splice variant in zebrafish and cell culture. This splice variant HIF-3α2 binds β-catenin independently of oxygen availability to destabilise the nuclear β-catenin complex. A detailed analysis uncovers that β-catenin destabilisation by HIF-3a2 occurs independently of its hypoxia response element (HRE) dependent transcriptional activity. In the zebrafish, the authors find HIF-3α2 expressed in all adult tissues analysed. Performing gain and loss of function experiments the authors state that HIF-3α2 is required for left-right asymmetry of the heart due to its repressive function on the formation of the Kupffer's vesicle.

This manuscript by Zhang and co-workers contains some interesting and well carried out in vitro work adding to the already complex world of HIF proteins. However, I have major concerns regarding the physiological role of HIF-3α2.

The authors state that HIF-3α2 mRNA overexpression results in general growth retardation and developmental delays (paragraph two, Discussion). They also show varying morphological phenotypes with most embryos looking pretty unhealthy (Figure 3—figure supplement 1). Their only marker analysis on embryos injected with mRNA of unknown concentration (200-800 pg as stated in the Experimental Procedures is very vague) show an early loss of somitic structures (Figure 3). The KV dependent establishment of brain and organ laterality during early embryonic development is a highly sensitive process and is influenced by numerous structures, processes and conditions ranging from cilia formation and motor protein function to gastrulation defects and even temperature (i.e. Basu and Brueckner, 2008; Bisgrove et al., 2005; Capdevilla et al., 2000; Dreosti et al., 2014; Nonaka et al., 1998; Supp et al., 1997; Wilkinson et al., 2009 and others). Given all the alterations observed by the authors, the specificity of the effect on gene expression in the Kupffer's vesicle and its connection to the concomitant rather mild effect on heart laterality needs a vigorous validation. Granted, the in vivo rescue experiments in Figure 4 regarding general early morphology look convincing, but canonical Wnt signaling has many important roles during early embryonic development. Therefore a minimum requirement to verify a potential specific effect would be to for instance manipulate gene expression specifically in the DFCs (gain and loss of function) or the KV as has been reported previously (e.g. Caron et al., 2012) and to perform a detailed analysis of KV morphology and function (for instance cilia morphology and functionality). Thereafter, the additional analysis of asymmetric organs other than the heart such as liver, pancreas and brain should be considered to validate the laterality defect.

The high mortality rate of HIF-3α2 knock out fish (subheading “C RISPR/Cas9-mediated editing of the hif-3α2 locus and ablation of Hif-3α2 increase Wnt/β-c atenin signalling”) emphazises the critical role of this gene during development in processes other than heart laterality, although the cause of the embryonic lethality has not been investigated. The morphological ko-phenotype shown in Figure 5—figure supplement 2 looks like a delayed development phenotype and has not much in common with a canonical Wnt phenotype. The same applies to the HIF-3α2 overexpression phenotype.

The authors' in vivo analysis concentrates on gain of function experiments, which is good because both the morpholino and CRISPR approach will affect also HIF-3α1 gene function if I am not mistaken. This should also be considered by the authors or explained in a more precise way.

Reviewer #2:

This paper describes the functional analysis of a HIF3a2 isoform and finds that it may be involved in downregulating wnt signalling.

Much of the data is convincing, however if I take a step back, a lot of data hinges on overexpression data which, just like morpholino data, I find it hard to be convinced by. To me the gold standard of proof of functional relevance and importance of a protein function is loss of function studies via genetic mutations.

I was pleased to see that a mutant had been made. However it was unclear to me what this mutant actually does molecularly, but this may be simply unclear presentation of the data. Importantly, the difference in intron exon structure is not explained properly, "it lacks portions of intron 1 and 11", this is unclear as spliced RNA has no introns. The figure is too small to get any wiser. Therefore, the paper needs to state the difference between HIF3a1 and HIF3a2 more clearly and explicitly somewhere in the main text; where is the alternative start codon does it correspond to a codon in the a1 isoform? In addition it needs to explain as well that the mutant obtained does NOT affect HIF3a1 and why. A DNA alignment of the 2 variant transcripts with annotated start codon and the deleted sequence in the mutant allele would do all of this. As sequence deletions in the primary transcript might affect splicing of other splice variants a northern blot might be the best way to prove specificity of the mutant.

The loss-of-function data are the make or break of this paper, and making and analysis of the mutant should not be in an additional figure. They should be at the center of the paper and convince me. qPCR is a nice technique to analyse gene expression differences in mutants, but dangerous as well, can general defects lead to increased gene expression?

I would be more convinced if a specific mutant phenotype is shown. For instance:

– Can known "wnt overexpression" phenotypes be identified, like a masterblind-type phenotype? There are various ISH markers that could be used to make such a phenotype visible. If there is no such phenotype, all is not lost….

– If this is the case, are the HIF3a2 mutants nevertheless interacting with wnt overexpressing mutants? I.e. can they enhance them, can you sort something based on a sensible phenotype and then genotype to show enrichment for the HIF3a2 mutants?

– Another way of doing this might be: if a clutch of embryos which segregates for the mutant is treated with low doses of wnt activators (e.g., Li), can the mutants be identified in a blinded sorting experiments?

The phenotype of the mutant is mysterious: early lethal or viable? That does not make sense, and therefore needs to be treated with caution.

Only a single allele is presented, how do we know it is clean? A second allele needs to be made and transheterozygotes need to show the same phenotype. If the allele is viable, is there a stronger and clearer maternal-zygotic phenotype?

Reviewer #3:

The manuscript by Zhang et al., entitled "An oxygen-insensitive Hif-3α isoform regulates left-right asymmetry be destabilizing the nuclear β-catenin complex", carries on the challenging, but highly important, work by this research team aiming to unravel the complexities of HIF-3α biology. Research on the O2 labile HIF-α subunits has been dominated by the relatively less complicated HIF1A and HIF2A genomic loci, which encode alternative mRNA species, but are not nearly as complex as human HIF3A which encodes > 19 predicted mRNA variants, and at least 8 distinct polypeptides. As such, the HIF-3α subfamily and its role in HIF biology should clearly be delineated for the sake of completeness. The manuscript reports on a novel Hif-3α variant in zebrafish ("Hif-3α2") and its possible role in regulating developmental left-right asymmetry. The authors wisely employ CRISPR/Cas9 genome editing to complement their Hif-3α overexpression studies, and conclude that Hif-3α imposes its developmental activity via inhibition of the Wnt/β-catenin signaling pathway. Here Hif-3α2 (which lacks bHLH, PAS, and ODD domains) binds β-catenin and destabilizes nuclear β-catenin complexes. Surprisingly this effect is independent of the well-characterized GSK3B pathway and O2 availability, as Hif-3α2 abundance is not regulated by O2 levels. The paper describes novel, O2-independent HIF functions, and suggests that Wnt/β-catenin can be regulated by previously uncharacterized mechanisms.

Major concerns:

1) The paper includes a substantial amount of well-controlled experiments, and appropriately complements the Hif-3α overexpression studies with CRISPR-mediated loss of function analyses in zebrafish. However, the Hif-3α2-/- fish exhibit significant mortality rates without showing the degree of morphological changes that might be expected if the deletants had the converse phenotypes detailed for the overexpression (i.e. in vivo gain of function) studies. Can the authors account for this? Is it possible Hif-3α2 effects on Wnt/β-catenin signaling are relatively minor (albeit detectable) compared to other in vivo activities?

2) The qRT-PCR data in Figure 1 do not mirror the RT-PCR amplicons depicted in Figure 1—figure supplement 1. For example, it is hard to detect an amplicon in the adult kidney, whereas the qRT-PCR data indicate that this is the tissue exhibiting high levels of Hif-3α2 transcripts.

3) Data shown for both the DFC marker gene, sox17, and number of migrating DFCs, while not statistically significant, suggest experimental trends and should be acknowledged as such.

4) Figure 4—figure supplement 1 has a labeling error that needs to be corrected so that the data can be interpreted properly.

5) The direct interaction data shown in Figure 6—figure supplement 3A suggesting β-catenin binds to GST-Hif-3α2, is very modest. Does this indicate a physical association that involves other factors beyond direct binding between the two proteins? Or perhaps in vivo post translational modifications needed?

6) Data shown in Figure 6 indicate that while MG132 treatment results in elevated levels of LEF1, co-expression with Hif-3α2 still resulted in a proportionate reduction in LEF1 protein accumulation. The comment "MG132 also inhibited the Hif-3α2-induced reduction in LEF1 levels" therefore appears to be inaccurate.

7) The statement "Mutant M2 completely lost its β-catenin inhibitory activity" is not consistent with data shown in Figure 7.

[Editors’ note: what now follows is the decision letter after the authors submitted for further consideration.]

Thank you for resubmitting your work entitled "An oxygen-insensitive Hif-3α isoform inhibits Wnt signaling by destabilizing the nuclear β-catenin complex" for further consideration at *eLife*. Your revised article has been favorably evaluated by Charles Sawyers (Senior editor), a Reviewing editor, and two reviewers. The manuscript is substantially improved but are just a couple of small points from the reviewers that need to be addressed before acceptance, as outlined below:

1) Please use the name hif-3αΔ42 instead of hif-3α2 mutant. (Comment from reviewer: It is courageous to state that hif-3α1 function is unaffected in hif-3αΔ42 mutants. This notion is likely with respect to the experimental data presented, but there may still be not identified functional changes due to for instance protein folding or other consequences of the deletion. While I feel that these potentialities do not need further addressing in the current manuscript, hif-3αΔ42 mutants carry a mutation in both hif-3α2 and hif-3α1 and possibly other existing hif-3α forms even if only hif-3α2 function seems affected. Even if this may be semantic, sticking to the naming hif-3αΔ42 instead of hif-3α2 mutant would be scientifically sound.)

2) How was the DFC cluster size determined? It would be helpful to have this mentioned in Figure 4 or its legend.

---

## [Author Response]

[Editors’ note: the author responses to the first round of peer review follow.]

We are very appreciative of the comments and constructive criticisms. We feel that we already have the additional data and/or explanations to address the three major issues raised by the reviews.

Issue #1: the HIF-3 isoform specific mutant fish.

This is arguably the biggest concern. This concern is mostly due to our poor presentation of the data. We have clearly failed to explain the nature of our mutant well enough. This mutant fish line was generated to eliminate the function of the Hifa2 isoform specifically. We have strong evidence that while the Hifa2 mRNA and protein levels were reduced to non-detectable levels, the expression levels of the full-length Hifa1 isoform did not change. More importantly, we will include new data to demonstrate that the mutation did not affect the functionality of the full-length Hifa1 isoform. We will include all these data and re-write this section to give an explicit clarification of the effects of our mutation on the different Hif3 isoforms.

We are also glad to report that we have obtained an additional mutant fish line. In this new mutant line, both the full-length Hifa1 and the oxygen-independent Hifa2 were targeted. Since the full-length Hifa1 protein is degraded under the normal oxygen condition, this new mutant line can be used to provide additional evidence for our hypothesis.

We would also like to clarify that the morpholino used was specific for the oxygen-independent Hifa2 isoform. It is our fault again that we did not make it explicitly clear. We will make this point very clear in the revised submission.

Issue #2: the left-right asymmetry phenotype.

We have used additional maker genes (spaw and lefty) and confirmed the effect of Hifa2 overexpression on the left-right asymmetry. We have also demonstrated this action of Hif3a2 in the LiPan reporter fish, in which liver is labeled by the expression of DsRed and pancreas by GFP expression). Importantly, overexpression of the full-length Hifa1 protein had no such activity, while the full-length Hifa1 protein was active in reducing body growth.

Issue #3: the lack of obvious phenotype in the mutant fish.

We agree that this is a very surprising result. A new study published last week by Dr Didier Stainier's lab (Rossi et al., Nature. 2015 Aug 13;524(7564):230-3. doi: 10.1038/nature14580) showed that in stable knockout of genes in zebrafish activated compensatory mechanisms to buffer against deleterious mutations while such genetic compensation was not observed after translational or transcriptional knockdown. This study is consistent with what we observed. We found that injection of aHifa2 isoform-specific morpholino resulted in phenotypes consistent with increased Wnt signaling activity. We have now performed more rigorous analyses of the phenotypes of the morphants and stable genetic mutant fish.

*As you will see, all three reviewers found merit in your study and recognised that this is an important and complex field of study. However, at present there are too many concerns to make your manuscript suitable for publication in* eLife *without major revisions that would be too substantial for what we require at* eLife*. In particular, it was felt that explicit clarification of the effects of your mutation on the various HIF3 isoforms was needed […]*

This concern was caused mostly by our poor presentation of the data in the old figures. The mutant fish line (*Δ42*) was generated to genetically delete the expression and function of Hif-3α2 specifically. It is our fault that we did not make it explicitly clear. We have re-worked the figure and revised the text to clearly explain the nature of the mutation (Figure 5 and Figure 5—figure supplement 1 and Figure 5—figure supplement 2). We have provided data showing that the Hif-3α2 mRNA and protein levels were undetectable in the mutant embryos (Figure 5). In contrast, the expression and the functionality of the full-length Hif-3α1 remained unchanged (Figure 5 and Figure 5—figure supplement 1). We have also obtained another allele (*Δ20)*. Both the full-length Hif-3α1 and Hif-3α2 were disrupted in *Δ20 embryos (*Figure 5—figure supplement 2). Since the full-length Hif-3α1 protein is degraded under normoxia, this new mutant line can provide additional evidence for our hypothesis. I would also like to clarify that the morpholino used was designed to specifically knockdown Hif-3α2. We have modified the relevant figure and re-written the text to give an explicit clarification (Figure 7 and Figure 7—figure supplement 1).

*[…] together with a much more rigorous analysis of the mutant phenotype, particularly in relation to Wnt signalling.*

We have performed additional analyses. The *Δ42* null mutant embryos had significantly elevated mortality between 6-9 hours post fertilization (hpf) (Figure 6 and Figure 6—figure supplement 1). This increase in mortality was also observed in *Δ20* embryos (Figure 5—figure supplement 2). Most of the surviving mutant embryos, however, looked morphologically normal and continued to develop (Figure 6). In contrast, no mortality increase was detected in the morpholino knocked-down embryos but the morphants displayed phenotypes characteristic of excessive Wnt signaling (Figure 7). An increase in maternal Wnt signaling was observed in the hif-3α2-/- mutant embryos but absent in the morphants (Figure 8). Further analyses revealed that the hif-3α2-/- mutant embryos displayed a transient and dynamic activation of zygotic Wnt signaling, while a more sustained increase of Wnt signaling was observed in the morphants (Figure 8). A recent study from Dr. Didier Stainier's lab (Rossi et al., Nature. 2015 Aug 13;524:230-3) showed that stable knockout of genes in zebrafish can activate compensatory mechanisms to buffer against deleterious mutations, while such genetic compensation was not observed after translational or transcriptional knockdown. This notion is consistent with what we observed. In support of this view, we have detected a concomitant reduction in BMP signaling in the hif-3α2-/- mutant embryos but not in the morphants (Figure 8). Following reviewer 2's suggestion, we treated mutant embryos with two *Wnt activating agents* Lithlium and BIO. The results showed that the *mutant embryos were more sensitive to* Lithlium and BIO (Figure 8). These data are consistent with the view that some compensatory mechanisms may have been activated in the genetic mutant to buffer against deleterious genetic mutation of hif-3α2-/-.

*The effects on left-right asymmetry, mentioned in the title, were not fully convincing and needed additional support.*

We have analyzed the expression of two additional asymmetrical maker genes (spaw and lft2) and confirmed the effect of Hif-3α2 overexpression on LR asymmetry (Figure 3). We have also investigated this action of Hif-3α2 using the LiPan reporter fish, in which liver is labeled by the expression of DsRed and pancreas by GFP expression (Figure 3). Importantly, overexpression of the full-length Hif-3α1 had no such effect (Figure 3). Because the central point of our study is on inhibition of canonical Wnt signaling by an oxygen-insensitive Hif-3α isoform and since the genetic mutant embryos showed little LR phenotype, we have changed the title into "An oxygen-insensitive Hif-3α isoform inhibits Wnt *signaling by destabilizing the nuclear β-catenin complex"*.

*Reviewer #1: […] Therefore a minimum requirement to verify a potential specific effect would be to for instance manipulate gene expression specifically in the DFCs (gain and loss of function) or the KV as has been reported previously (e.g. Caron et al., 2012) and to perform a detailed analysis of KV morphology and function (for instance cilia morphology and functionality). Thereafter, the additional analysis of asymmetric organs other than the heart such as liver, pancreas and brain should be considered to validate the laterality defect.*

a) In pilot experiments, we tested different doses (200 pg to 600 pg) of Hif-3α2 mRNA. For the data reported in this manuscript, the injected mRNA dose was 800 pg for Hif-1α’, Hif-3α1’, GFP mRNA, and 600 pg for Hif-3α2 mRNA. This information is now added to the figure legends.

b) We observed two major phenotypes resulted from Hif-3α2 overexpression. One is the global growth retardation and developmental timing delay. This action apparently requires the transcriptional activity because it was abolished by deletion the TAD domain. Another phenotype is the LR asymmetry defect. This action is unique to Hif-3α2 because no such phenotype was observed in the Hif-3α1 expressing embryos. This action does not appear to require the HRE-dependent transcriptional activity (TA) because the TA dead mutant, A36G mutant, caused a similar heart randomization.

c) We have added additional data on spaw and lft2 expression patterns (Figure 3). We have also added new data on liver and pancreas location using the LiPan reporter fish line (Figure 3).

d) We agree that the KV dependent establishment of brain and organ laterality is a highly sensitive process and influenced by numerous factors. We showed that overexpression of Hif-3α2 altered LR asymmetry, while the full-length Hif-3α1 protein had no such activity. We examined DFC clustering and migration and found no significant change (Figure 4). Wnt signaling plays several distinct roles in vertebrate axis specification. While maternal Wnt/β-catenin signaling contributes to the dorsoventral axis establishment in early embryogenesis, the zygotic Wnt/β-catenin signaling is involved in anteroposterior axis specification. It has been suggested that these two roles are mediated by two distinct sets of target genes. Canonical Wnt/β-catenin signaling also regulates KV development and the establishment of the LR axis in zebrafish. Overexpression of Hif-3α2 mainly impairs KV development and results in subsequent changes in organ laterality (Figure 3). This is probably because it takes some time for the injected mRNA to be translated into protein and for resulting biological changes to manifest. Since Hif-3α2 mRNA is maternally deposited, the endogenous Hif-3α2 may act earlier. Indeed, we observed a significant increase in maternal Wnt signaling and early mortality in the hif-3α2-/- embryos (Figure 6 and Figure 8). Since the goal of our study is to determine the role of the oxygen-insensitive Hif-3α2 in regulating Wnt signaling, we did not perform the suggested experiments to manipulate gene expression specifically in the DFCs or the KV. As mentioned earlier, we have modified the title of this manuscript to tone down the emphasis on LR asymmetry development. Another and more practical reason is that the first author of this work has moved on. Given that this manuscript already contains 6 years of work and 10 multi-panel figures plus 14 supplemental figures, we feel that this body of new and original data is sufficient to stand on its own. We sincerely hope the reviewer would agree.

*The high mortality rate of HIF-3α2 knock out fish (subheading “C RISPR/Cas9-mediated editing of the hif-3α2 locus and ablation of Hif-3α2 increase Wnt/β-c atenin signalling”) emphazises the critical role of this gene during development in processes other than heart laterality, although the cause of the embryonic lethality has not been investigated. The morphological ko-phenotype shown in Figure 5—figure supplement 2 looks like a delayed development phenotype and has not much in common with a canonical Wnt phenotype. The same applies to the HIF-3α2 overexpression phenotype.*

We agree that the elevated mortality rate is important. We have more carefully analyzed this phenotype. Six pairs of mutant fish were bred 4 times. The mortality rate varied considerably among different pairs of mutant fish and between different breeding cycles in the same pair (Figure 6—figure supplement 1), but was significantly higher than that of the wild type controls (Figure 6). This increase in embryo mortality was also observed in *Δ20* embryos (Figure 5—figure supplement 2). Most of the surviving mutant embryos, however, looked morphologically normal and continued to develop. In contrast, no mortality increase was detected in the morphants. The morphants displayed phenotypes characteristic of excessive Wnt signaling (Figure 7). The MO effect was unlikely due to toxicity because the Hif-3α2 targeting MO did not increase embryo mortality, nor did it increase p53 mRNA expression at the injected doses (Figure 7—figure supplement 1). Furthermore, the hif-3α*Δ42* embryos were less sensitive to Hif-3α2 MO (Figure 7). These data are consistent with the notion that stable knockout of hif-3α2 may have activated compensatory mechanisms to buffer against deleterious mutations.

*Reviewer #2:*

*[…] Much of the data is convincing, however if I take a step back, a lot of data hinges on overexpression data which, just like morpholino data, I find it hard to be convinced by. To me the gold standard of proof of functional relevance and importance of a protein function is loss of function studies via genetic mutations. I was pleased to see that a mutant had been made. However it was unclear to me what this mutant actually does molecularly, but this may be simply unclear presentation of the data. Importantly, the difference in intron exon structure is not explained properly, "it lacks portions of intron 1 and 11", this is unclear as spliced RNA has no introns. The figure is too small to get any wiser. Therefore, the paper needs to state the difference between HIF3a1 and HIF3a2 more clearly and explicitly somewhere in the main text; where is the alternative start codon does it correspond to a codon in the a1 isoform? In addition it needs to explain as well that the mutant obtained does NOT affect HIF3a1 and why. A DNA alignment of the 2 variant transcripts with annotated start codon and the deleted sequence in the mutant allele would do all of this. As sequence deletions in the primary transcript might affect splicing of other splice variants a northern blot might be the best way to prove specificity of the mutant.*

Following this suggestion, we have re-worked the figure and re-worded the text to clearly explain the nature of the mutation (Figure 5 and Figure 5—figure supplement 1). A second allele was also obtained and used in this study (Figure 5—figure supplement 2).

*The loss-of-function data are the make or break of this paper, and making and analysis of the mutant should not be in an additional figure. They should be at the center of the paper and convince me. qPCR is a nice technique to analyse gene expression differences in mutants, but dangerous as well, can general defects lead to increased gene expression?*

*I would be more convinced if a specific mutant phenotype is shown. For instance:*

– Can known "wnt overexpression" phenotypes be identified, like a masterblind-type phenotype? There are various ISH markers that could be used to make such a phenotype visible. If there is no such phenotype, all is not lost…

*– If this is the case, are the HIF3a2 mutants nevertheless interacting with wnt overexpressing mutants? I.e. can they enhance them, can you sort something based on a sensible phenotype and then genotype to show enrichment for the HIF3a2 mutants?*

*– Another way of doing this might be: if a clutch of embryos which segregates for the mutant is treated with low doses of wnt activators (e.g., Li), can the mutants be identified in a blinded sorting experiments?*

We thank this reviewer for this criticism and helpful suggestion. We have performed additional experiments. The *Δ42* null mutant embryos had significantly elevated mortality in early stages. Among the surviving mutant embryos, a few did show reduced forebrain and smaller eyes (Figure 6). There was a transient increase in both maternal and zygotic Wnt target genes in the surviving and normal looking embryos (Figure 6 and Figure 8). Expression of flh, a mid/caudal diencephalon marker gene, increased at 6 hpf but returned to the control levels at 10 hpf (Figure 6). Following this reviewer's suggestion, we have treated the embryos with Lithium and BIO. A significantly greater portion of the hif-3αΔ42 embryos exhibited the no eye or small eye phenotype compared to the wild type controls (Figure 8). We further showed that knockdown of Hif-3α2 significantly increased Wnt/β-catenin activity and resulted in phenotypes resembling those of mbl mutant embryos ((Figure 7). These and other data led us to postulate that stable knockout of hif-3α2 has activated compensatory mechanisms to buffer against deleterious mutations.

*The phenotype of the mutant is mysterious: early lethal or viable? That does not make sense, and therefore needs to be treated with caution.*

*Only a single allele is presented, how do we know it is clean? A second allele needs to be made and transheterozygotes need to show the same phenotype. If the allele is viable, is there a stronger and clearer maternal-zygotic phenotype?*

Please see response to reviewer 1's comment #2.

Reviewer #3:

*1) The paper includes a substantial amount of well-controlled experiments, and appropriately complements the Hif-3α overexpression studies with CRISPR-mediated loss of function analyses in zebrafish. However, the Hif-3α2-/- fish exhibit significant mortality rates without showing the degree of morphological changes that might be expected if the deletants had the converse phenotypes detailed for the overexpression (i.e.* in vivo *gain of function) studies. Can the authors account for this? Is it possible Hif-3α2 effects on Wnt/β-catenin signaling are relatively minor (albeit detectable) compared to other* in vivo *activities?*

We agree that the lack of LR axis phenotype was surprising. We have performed additional experiments with this mutant line (*Δ42*) and another mutant line (*Δ20*). In both mutant lines, there were significant increases in Wnt signaling (Figure 5 and Figure 5—figure upplement 1 and Figure 5—figure supplement 2). We also compared the genetic mutants with MO-knocked down embryos (please see responses to editor's comment#2 and Reviewer 2's comments). These data suggested that some compensatory mechanisms have been activated in the genetic mutant to buffer against deleterious genetic mutations of hif-3α2-/-. However, it is also possible that Hif-3α2 may have other activities in vivo. In fact, our results show that Hif-3α2 can regulate gene expression via its TAD domain (Figure 2) and this activity is required for its role in body growth (Figure 3—figure supplement 1). We cannot exclude the possibility Hif-3α2 may affect other pathways directly or indirectly. We have discussed these possibilities (paragraph three, Discussion).

*2) The qRT-PCR data in Figure 1 do not mirror the RT-PCR amplicons depicted in Figure 1—figure supplement 1. For example, it is hard to detect an amplicon in the adult kidney, whereas the qRT-PCR data indicate that this is the tissue exhibiting high levels of Hif-3α2 transcripts.*

We have deleted the redundant RT-PCR data, which were not quantitative.

*3) Data shown for both the DFC marker gene, sox17, and number of migrating DFCs, while not statistically significant, suggest experimental trends and should be acknowledged as such.*

We have made this point more clear.

*4) Figure 4—figure supplement 1 has a labeling error that needs to be corrected so that the data can be interpreted properly.*

We have modified the figure.

*5) The direct interaction data shown in Figure 6—figure supplement 3A suggesting β-catenin binds to GST-Hif-3α2, is very modest. Does this indicate a physical association that involves other factors beyond direct binding between the two proteins? Or perhaps in vivo post translational modifications needed?*

The direct interaction between β-catenin and GST-Hif-3α2 is highly reproducible. We have replaced the old image with an image from another experiment (Figure 9—figure supplement 2). We cannot exclude the possibility that other factors may be involved in this physical association. Post-translational modification is also plausible. These possibilities have been added to the Discussion (paragraph five).

*6) Data shown in Figure 6 indicate that while MG132 treatment results in elevated levels of LEF1, co-expression with Hif-3α2 still resulted in a proportionate reduction in LEF1 protein accumulation. The comment "MG132 also inhibited the Hif-3α2-induced reduction in LEF1 levels" therefore appears to be inaccurate.*

We have changed this sentence to the following: "MG132 treatment also partially inhibited the Hif-3α2-induced reduction in Lef1 levels ".

*7) The statement "Mutant M2 completely lost its β-catenin inhibitory activity" is not consistent with data shown in Figure 7.*

We agree. This sentence has been changed to: "Mutant M2 had partial β-catenin inhibitory activity".

[Editors’ note: what now follows is the decision letter after the authors submitted for further consideration.]

1) Please use the name hif-3αΔ42 instead of hif-3α2 mutant. (Comment from reviewer: It is courageous to state that hif-3α1 function is unaffected in hif-3αΔ42 mutants. This notion is likely with respect to the experimental data presented, but there may still be not identified functional changes due to for instance protein folding or other consequences of the deletion. While I feel that these potentialities do not need further addressing in the current manuscript, hif-3αΔ42 mutants carry a mutation in both hif-3α2 and hif-3α1 and possibly other existing hif-3α forms even if only hif-3α2 function seems affected. Even if this may be semantic, sticking to the naming hif-3αΔ42 instead of hif-3α2 mutant would be scientifically sound.)

We agree with the reviewers. We have changed the mutant fish name from “hif-3α2 mutant” into “hif-3αΔ42” throughout the revised manuscript.

2) How was the DFC cluster size determined? It would be helpful to have this mentioned in Figure 4 or its legend.

The DFC cluster size was determined by measuring sox17 mRNA expression domain using ImageJ. The values are normalized by those of the GFP mRNA injected control group. This information is now added to Figure 4 legend.